# Poverty reduction in rural China: Does the digital finance matter?

**Boou Chen** [ORCID][⊕], **Chunkai Zhao** *[⊕]

Institute of Finance and Economics Research, Shanghai University of Finance and Economics, Shanghai, China

⊕ These authors contributed equally to this work.
* ckz_zhao@163.com

## Abstract

As digital finance is widely spread and applied in China, this new format of financial technology could become a new way to reduce poverty in rural areas. By matching digital financial indexes of the prefectural-level cities with microdata on rural households from the China Household Finance Survey (CHFS) in 2017, we find that digital finance significantly suppresses absolute poverty and relative poverty among rural households in China, which is supported by a series of robustness tests, such as the instrumental variable approach, using alternative specifications, and excluding extreme observations. Additionally, we provide evidence that the poverty reduction effect of digital finance is likely to be explained by alleviating credit constraints and information constraints, broadening social networks, and promoting entrepreneurship. Our findings further complement the research field on financial poverty reduction and offer insights for the development of public financial policies of poverty reduction in other countries, especially in some developing countries.

## 1. Introduction

Poverty reduction is the basis for maintaining social stability and has become one of the major challenges in developing countries. China is the largest developing country in the world and once had the largest rural poor population [1]. China has made great efforts to solve the problems of poverty and implemented a series of poverty reduction measures in different stages. Before 1978, the primary objective of antipoverty was to ensure basic survival of farmers, and the main measures were low-level social assistance together with mutual aid and cooperation [2]. However, in 1978, according to the rural poverty standard calculated at the price level of that year, 770 million people are still in absolute poverty, accounting for 97.5% of the rural population. From 1978 to 2012, China's institutional reform had significantly relieved the poverty in rural areas, more than 700 million people in rural China overcame the problems of poverty. In 2013, the Chinese government implemented the targeted poverty alleviation (TPA). The TPA ensured that assistance accurately reaches poverty-stricken villages and households, and combined five approaches to eliminate poverty, which are industrial development, resettlement, ecological compensation, strengthened education and social security [2–5]. The latest

**Data Availability Statement:** All relevant data are within the paper.

**Funding:** This work was supported by the Fundamental Research Funds for the Central

Universities for Shanghai University of Finance and Economics (No.QCDC-2020-10; No. QCDC-2020-21), and the Graduate Innovation Fund of Shanghai University of Finance and Economics (No. CXJJ-2019-434; No. CXJJ-2020-304).The funders had no role in study design, data collection and analysis, decision to publish, or preparation of the manuscript.

**Competing interests:** The authors have declared that no competing interests exist.

report from the China's National Bureau of Statistics shows that from 2012 to 2019, the average annual reduction rate of rural poverty was as high as 51.06%, and problem of absolute poverty was completely solved in 2020. However, the relative poverty of rural households remains severe due to the large disparity between urban and rural development in China [6, 7].

Among many poverty reduction approaches, the effectiveness of financial poverty alleviation has always been concerned. In terms of the macro-economic, financial development may shrink poverty through economic growth, urbanization, industrialization, and international trade [8–16]. From the micro perspective, financial development may reach more low-income groups and reduce the incidence of relative poverty, especially as countries increasingly focus on inclusive financial development [17–23]. In recent years, digital finance has received widespread attention as financial development and the Internet have become more and more closely integrated.

Digital finance is a new financial format that relies on the Internet and information technology tools to carry out financial services and benefit more groups [18, 20, 24, 25]. In essence, it is an important type and application of Financial Technology (FinTech) [26]. China's digital finance is mainly mobile payments, online loans, digital insurance and online investments [25–27]. With the spread of the Internet and smartphones, digital finance in China has made great strides, which has greatly increased the accessibility and convenience of formal financial services, especially for those who previously did not have access to them [28–30]. However, since research on the impact of digital finance on poverty reduction is still very limited, we try to explore the role of digital finance in China's rural poverty reduction, as China is the most widely used country for digital finance in the world.

The role of digital finance has been noted by many scholars. On the one hand, they found that digital finance not only promotes economic growth, but also plays a positive role in reducing the rural-urban gap [31]. On the other hand, in terms of the impact on individuals and households, the functions of digital finance can be attributed as: easing the financing constraints of low-income groups [32, 33], achieving consumption smoothing [20, 25, 30, 34], promoting the possibility of entrepreneurial activities [32, 35], and increasing the potential benefits of entrepreneurship [33, 36]. Additionally, few studies explored the impact of digital finance on poverty alleviation. Another literature similar to our study comes from Suri and Jack (2016), who obtained the conclusion that FinTech contributes to poverty reduction [37]. They found that M-Pesa, which is mobile banking service launched by mobile operator "Safaricom" in Kenya, enabled many Kenyan women to move out of subsistence farming and into small-scale enterprises to earn higher incomes by providing additional financial resources [37].

However, there is some controversy in the previous literature on the poverty reduction effect of FinTech. On the one hand, FinTech requires the use of the Internet or mobile devices, but some poor people may have a digital divide [38], making it difficult to realize the poverty alleviation benefits of digital finance [22]. On the other hand, poverty reduction effects of FinTech may be short-term [39], affected by the imperfection of credit and financial systems. Therefore, further exploration is still needed on whether digital finance can effectively alleviate poverty.

In this paper, we have some meaningful findings, which further complement the previous research field. First, although the role of finance in poverty reduction is widely recognized, little is known about the effects of digital finance on rural poverty reduction in China. By matching digital financial indexes of the prefectural-level cities and rural household microdata from the China Household Finance Survey (CHFS) in 2017, we find that digital financial significantly suppresses absolute and relative poverty among Chinese rural households.

Second, the abundant information in the data from the CHFS provides us with fertile ground to figure out the possible mechanisms by which digital finance affects the incidence of poverty. The unique Chinese setting helps us to thoroughly understand how digital finance has a stable positive impact on poverty reduction among Chinese rural households, which are easing credit constraints and information constraints, enhancing social networks, and promoting entrepreneurial activities. Moreover, these findings may provide some useful inspiration of poverty reduction for other developing countries that are similar to China.

Third, the results of heterogeneity analysis confirm the inclusiveness of digital finance, i.e., digital finance benefits more socially disadvantaged groups. We find that digital finance is more beneficial for older and uneducated rural households to escape poverty. Furthermore, our results further enrich the relevant literature on the inclusive finance and functions of digital finance [2, 17, 22, 28].

## 2. Digital finance in China

Digital finance in China started with the launch of Alibaba's Alipay in 2004. Until 2013, with the birth of the Internet financial product, Yu'E Bao, and the popularity of mobile payments, digital finance became known to a wider public [27]. Subsequently, driven by FinTech and mobile Internet technologies, China's digital financial system, represented by mobile payments, Internet wealth management, online crowdfunding, and online lending, was formally established. More importantly, the development of digital finance cannot be separated from the guidance and support of national policies. The standardized development of Internet finance or digital finance has been mentioned in the Chinese government work reports in all years. In addition, a series of policy documents, such as the Guidance on Promoting the Healthy Development of Internet Finance issued in 2015 led by the People's Bank of China and the Guidance on Promoting the Standardized and Healthy Development of the Platform Economy issued by the State Council in 2019, have played an irreplaceable role in promoting the digital finance development in China.

To date, China has become one of the best developed and most widely used countries in the world for digital finance [33]. The establishment and growth of a large number of digital finance companies has laid the foundation for the long-term and sustainable development of digital finance. According to the "2018 Fin Tech 100" released by KPMG International and H2 Ventures, there are three financial technology companies from China in the top five of this list: Ant Financial (1st), JD Finance (2nd) and Baidu Financial (4th). Furthermore, digital finance has injected new momentum into the city's economic development. Some digital financial center cities, such as Hangzhou, have established new city brands and ushered in new development opportunities with the digital finance.

According to the Digital Financial Inclusion Index (DFII) compiled by the Institute of Digital Finance of Peking University in collaboration with Ali Finance, we found some characteristics of digital finance development in China. First, as shown in Fig 1, from 2011 to 2018, digital finance has developed rapidly in China. Second, the differences in city-level DFII between regions are gradually converging in Fig 2 and the differences between regions are narrowing, which is consistent with the findings from Huang and Tao (2019) [27]. They found that the difference in DFII between the most and least developed regions of the Chinese economy has decreased from 50.4% in 2011 to 1.4% in 2018.

Digital finance in China has the remarkable feature of promoting financial inclusion. It not only provides financial services such as mobile payment, bill payment, deposit and loan to small and micro enterprises and low-income people in backward and remote areas [32, 33],

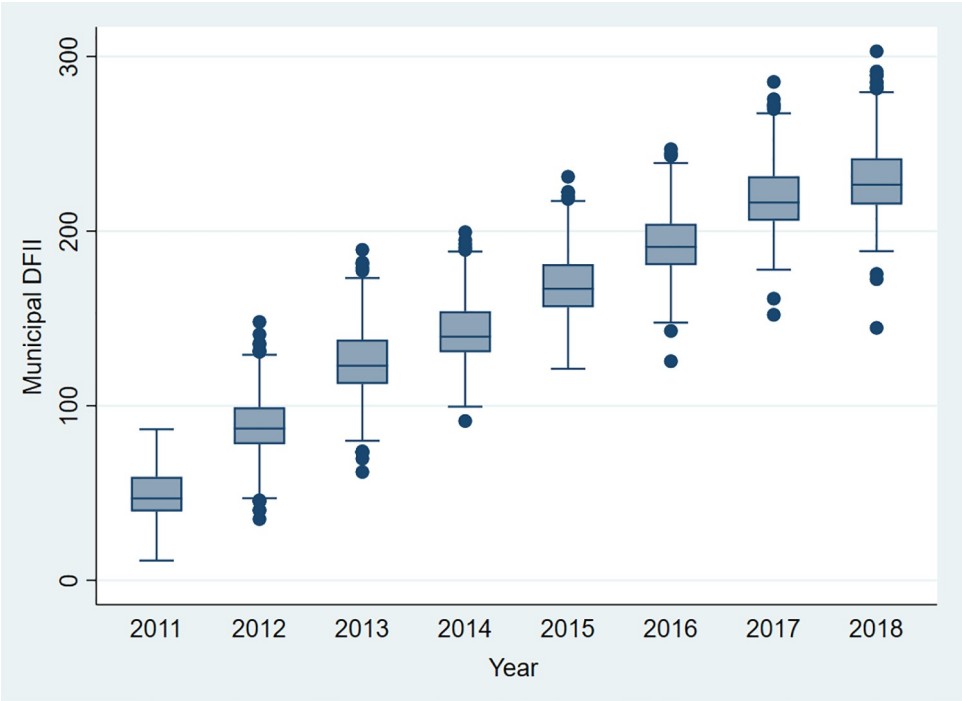

**Fig 1. The box-plot of municipal DFII in China from 2011 to 2018.**

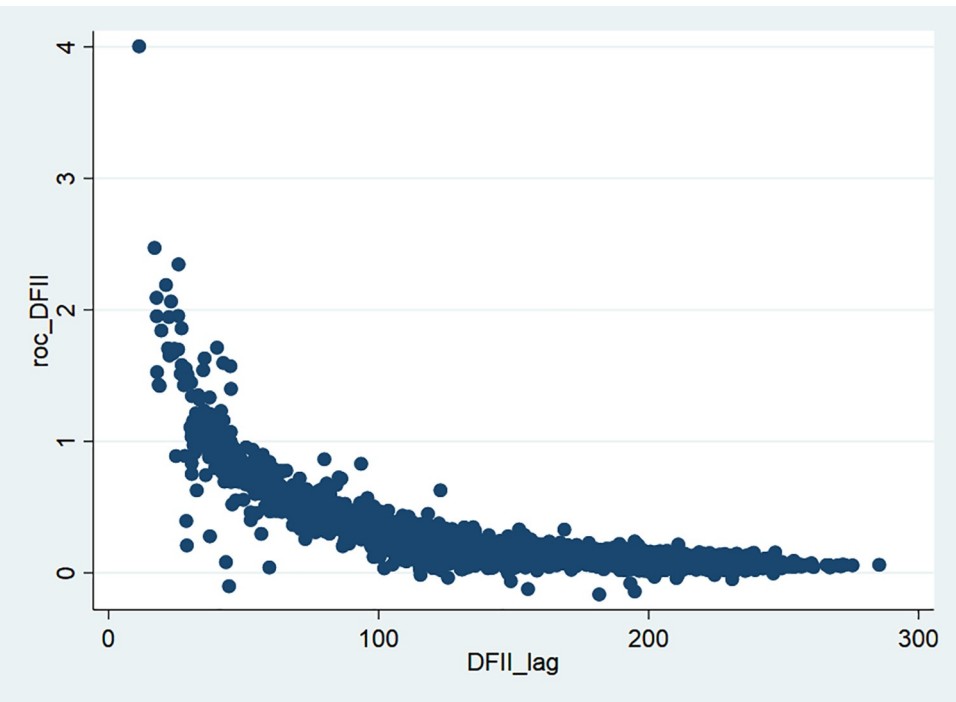

**Fig 2. The rate of change of city-level DFII in China from 2011 to 2018.**

but also enables them obtain formal financial with lower transaction costs and a more convenient way [25, 28].

## 3. Theoretical framework

In this section, we discuss the theoretical mechanisms of digital finance on poverty reduction among Chinese rural households. We classify possible mechanisms into the following categories: credit constraints, information advantages, social networks, and entrepreneurial activities.

### 3.1. Credit constraints

Digital finance may reduce the incidence of poverty by alleviating credit constraints. Low-income and poor rural households often have strong credit constraints and are affected by lack of access to the inadequate provision of financial services, making it difficult to improve their economic conditions [40]. Traditional financial institutions have high unit costs for granting agricultural credit and lower overall returns [41], while rural households live more dispersedly, and loans available to rural households and micro enterprises are often in a small scale. Therefore, poor rural households are difficult to achieve the formal financial services from traditional financial institutions, and unable to obtain additional and funds for production or other investments [42].

Compared to traditional financial institutions, digital finance only needs less investment for system construction and development at the initial stage, and can reduce the degree of information asymmetry and the risk of adverse selection by integrating a large number of online user information [36]. It further promotes the development of financial inclusion, and reduces the rate of financial exclusion among the poor. In addition, benefit from digital finance, loan application only needs to be completed on the Internet terminals, which is more convenient and friendly for the rural households with limited financial knowledge [20]. Digital finance helps poor rural households alleviate their credit constraints by increasing their possibilities of achieving financial services and simplifying the process of loan application. The alleviation of credit constraints on rural households may increase the family income and improve their ability to bear risks, which reduce the incidence of poverty [43]. Therefore, we put forward the first hypothesis:

**Hypothesis 1**: Digital finance may reduce rural poverty by alleviating credit constraints.

### 3.2. Information constraints

In addition to credit constraints, poor and low-income rural households also face strong information constraints. There is a clear "digital gap" with middle-income and high-income groups in the production, employment, and life for the poor. Some studies confirmed the impact of the digital divide on the income gap [44–46]. On the contrary, with the rapid development of information and communication technology (ICT), the use of smartphones and the Internet has a significant role in increasing individual income [47, 48]. Digital finance is a new financial format that combines the ICT with traditional financial services to reach more groups [20]. Therefore, the development of digital finance may further strengthen the role of ICT in narrowing the income gap and further promote poverty reduction by alleviating the information constraints of poor and low-income households.

Furthermore, low-income people usually lack financial knowledge and have limited ability to collect and identify data from the Internet. Therefore, although the development of ICT makes it easier to obtain information and reduces the cost of obtaining information, it may

still be difficult to benefit low-income groups. With the help of financial platforms and big data technology, digital finance will deliver information that is more useful to clients to improve their economic conditions and is more compatible with user characteristics [18, 35]. People can easily obtain information related to agricultural production and management, employment, finance and daily life timely from digital financial platforms [32, 33]. After big data analysis, this part of information is highly matched with users, more accurate and transparent. It may help to promote the employment and improve the efficiency of agricultural production [28], thus increase their income and reducing the incidence of poverty. In addition, even if the information received is only about daily life, rural households have more opportunities to reallocate resources optimally and improve their ability to cope with external risk shocks [24]. To sum up, we propose the second hypothesis:

**Hypothesis 2**: Digital finance is likely to curb rural poverty by leveraging information and alleviating information constraints.

### 3.3. Social networks

Digital finance may help rural households expand social networks and strengthen ties with relatives and friends. In China, social networks are important institutional social capital that could explain the role of digital financial development in alleviating rural household poverty. Previous literature suggested that social networks were closely related to individuals' income, employment, and occupation choices [49, 50]. In a typical relational society, social networks even play an important role in lifting rural Chinese families out of poverty [51, 52].

The digital finance has provided people with a more convenient way to pay and increased the frequency of social engagement. Relying on the Internet platform, digital finance provides people with an effective means of communication and social interaction. For example, WeChat Pay was developed by relying on WeChat, the largest online social platform in China. By combining the custom of WeChat red envelopes with traditional Chinese features, it has greatly enhanced the online social interaction experience [53]. Additionally, digital financial development has the potential to increase people's online accessibility and facilitate their participation in online social networking [54, 55]. Thus, we derive the third hypothesis:

**Hypothesis 3**: Digital finance is likely to reduce rural poverty by expanding social networks.

### 3.4. Entrepreneurial activities

Digital finance may alleviate poverty by promoting entrepreneurial activities of rural households. Entrepreneurial activities as a solution to reduce poverty has been explored by many research [56–59]. Entrepreneurship, especially informal entrepreneurship, as an important source of increasing household income in China, is an effective way to get rural households out of the poverty trap [57, 58]. However, strong credit constraints will hinder entrepreneurial behavior, especially for low-income and poor families [60–62]. The financing function of digital finance improves the credit availability of potential entrepreneurs [63], and has a positive impact on rural households' entrepreneurial activities [32]. With the help of digital financial platforms, entrepreneurial farmers can obtain a large amount of information related to entrepreneurship, and strengthen cooperation with buyers or other entrepreneurs, so as to evaluate accurately the feasibility and market prospects of entrepreneurial projects [35]. In addition, mobile payment can reduce transaction costs and make transactions more convenient and safer [64, 65]. The reduction of transaction costs and transaction risks increases the potential returns of entrepreneurs [36]. In summary, we formulate the fourth hypothesis:

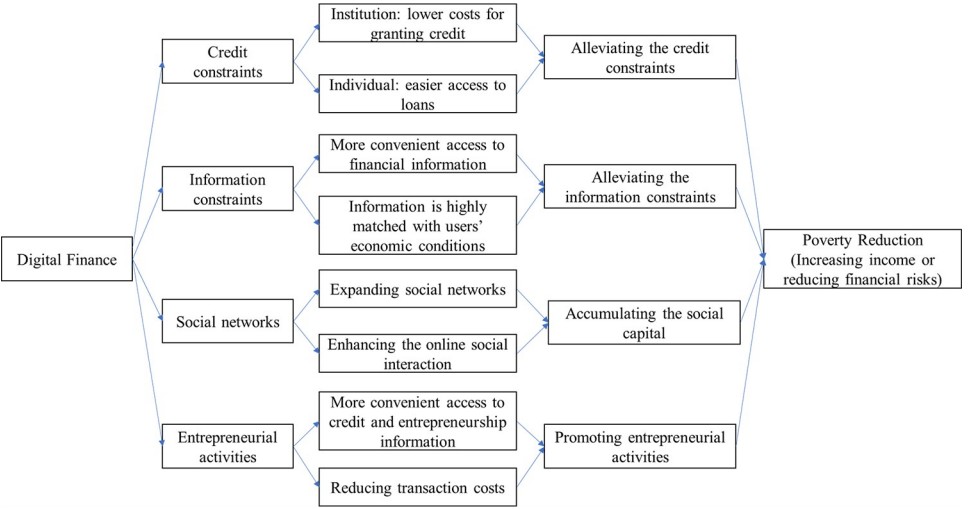

**Fig 3. The mechanisms of digital finance on poverty reduction.**

**Hypothesis 4**: Digital finance may alleviate poverty by promoting entrepreneurial activities of rural households.

Given the multiplicity of mechanisms through which digital finance reduces poverty, we provide Fig 3 to more clearly articulate these mechanisms. In addition, it is important to note that these mechanisms are not independent of each other, and it is very difficult to truly identify the role of each mechanism. Below using relevant survey questions in the CHFS involving these mechanisms, we attempt to empirically examine which hypotheses are more consistent with the data and provide some suggestive evidence for the role of these mechanisms in explaining the poverty reduction effects of digital finance.

## 4. Data, variables, and methodology

### 4.1. Data source

The microdata in this paper come from the fourth round of the China Household Finance Survey (CHFS), which was newly released by Southwest University of Finance and Economics in 2017. As a nationally representative household database, the CHFS data covers 29 provincial-level administrative regions, 228 cities (prefectures), and 609 villages in mainland China, excluding Tibet and Xinjiang, with microdata on 12,732 rural households. In addition to providing demographic characteristics, the CHFS also focuses on investigating household economic and financial information, such as household income, liabilities, assets, consumption, employment and entrepreneurship, and payment habits. In particular, in the work and income section, the CHFS records in detail the various household incomes, providing a good source of data for determining whether rural households are poor. The CHFS is one of the most widely used databases for studying poverty issues in China [21, 66, 67].

As mentioned earlier, the data on digital finance we use come from the DFII of the Institute of Digital Finance of Peking University. These indexes portray the development of digital finance in China at the provincial, city, and county levels, and are the main indicators currently used to explore the digital finance development in China [18, 20, 25, 28]. In the CHFS, since only the prefectural-level city where the household is located is disclosed, and not the district or county, we select city-level digital financial indicators. In addition, considering that the

CHFS survey was conducted in the first half of 2017, we used the 2016 Digital Finance Index for matching.

## 4.2. Variables

**4.2.1. Poverty.**   Similar to the previous literature [7, 67–70], our study draws on two measures of household poverty in rural China: absolute poverty and relative poverty. Starting in 1986, the Chinese government began setting the rural poverty line and using it as a standard for identifying the size of the rural poor and the incidence of rural poverty. China's first poverty alleviation standard, set at an annual per capita income of 206 yuan per year for farmers, was subsequently updated several times to reflect the change in CPI [67]. In 2016, the State Council Poverty Alleviation Office of China updated the absolute poverty line to 2,855 yuan. On a purchasing power parity basis, this standard was equivalent to $2.20 per day, slightly higher than the international extreme poverty standard of $1.90. Taken together, we define rural households with annual per capita income below 2855 yuan as absolutely poor and set it to 1, and 0 otherwise.

Regarding relative poverty, the definitions of international organizations and countries vary. For example, in 1976, the Organization for Economic Cooperation and Development (OECD) proposed 50% of the median income of a country or region as the poverty, which is widely used internationally. In addition, the World Bank defines relative poverty as having an income below 1/3 of the average income, and in Europe, the relative poverty rate is measured by the percentage of the population whose income level is below 60% of the median income. We mainly refer to the OECD, but we have reduced the ratio considering that the OECD is dominated by developed countries. Specifically, we consider those with household income per capita in the bottom 25% as relatively poor and set it to 1, and 0 otherwise.

**4.2.2. Digital finance.**   Our main explanatory variable is the 2016 digital finance aggregation index of the prefectural-level cities (one-period lagged), composing of three main sub-indicators, namely coverage breadth, usage depth, and digitization. Specifically, coverage breadth is mainly measured by the coverage of digital financial accounts and usage depth includes payments, money funds, lending (including consumer loans and micro and small business loans), insurance, investment, and credit. The indicator system of digital finance is shown in Table 1. Moreover, since elements such as mobility and convenience included in digitization are closely related to household consumption [20], we do not include them in our core explanatory variables.

**4.2.3. Control variables.**   We have included two levels of control variables in this paper. First, at the level of the characteristics of the head of household, considering the potential nonlinear effects, we choose *Age* and *Age squared* [21]. The remaining control variables include *Gender*, *Unschooled*, *Primary school*, *Junior middle school*, *Senior high school*, *Good health*, and *Poor health*, which we consider the positive effects of the good education, and self-rated health on household poverty reduction [67, 70, 71]. The highest education of the householder in the sample is junior college degree. To avoid collinearity problems, junior college and ordinary health are omitted from the education and self-rated health, respectively. The binary variable, *Good health*, is set to 1 when the self-rated health is "good" or "very good". Conversely, *Poor Health* is assigned a value of 1 when the self-rated health is "poor" or "very poor".

Second, household characteristics, such as *Ln consumption*, *Consumption-income ratio*, *Current deposit*, *Fixed deposit*, *Debt-income ratio*, *Child dependency ratio*, *Elderly dependency ratio*, *Housing ownership*, and *Car ownership* are taken into consideration, because of positive roles of household wealth and assets in poverty reduction [67, 72, 73]. Instead, child and elderly dependency ratio may be the key elements that contribute to rural household poverty [21, 74, 75].

**Table 1. The indicator system of digital finance.**

| | | |
|---|---|---|
| Coverage breadth | Alipay account coverage | Number of Alipay accounts per 10,000 people |
| | | Proportion of Alipay tied card users |
| | | Average number of bank cards tied to each Alipay account |
| Usage depth | Payment | Frequency of payments per capita |
| | | Amount of payments per capita |
| | | Percentage of active users with high frequency (50 times and above) |
| | Money funds | Frequency of purchasing Yu E Bao per capita |
| | | Amount of purchasing Yu E Bao per capita |
| | | Number of Alipay users who purchase Yu E Bao per 10,000 people |
| | Lending | Number of users with Internet consumer loans per 10,000 Alipay adult users |
| | | Number of loans per capita |
| | | Amount of loans per capita |
| | | Number of users with Internet micro and small business loans per 10,000 Alipay adult users |
| | | Number of loans per micro and small operators |
| | | Amount of loans per micro and small operators |
| | Insurance | Number of insured users per 10,000 Alipay users |
| | | Number of insurances per capita |
| | Investment | Number of Alipay users involved in Internet investment per 10,000 people |
| | | Number of Internet investment per capita |
| | | Amount of Internet investment per capita |
| | Credit | Number of credit calls per capita |
| | | Number of users employing credit-based services per 10,000 Alipay users |

**4.2.4. Descriptive statistics.** The definition and descriptive statistics of main variables are represented in Table 2, and all continuous variables (e.g., *Ln consumption*) are winsorized at the 1% level. After excluding missing values, our final observations include 11,816 rural households, of which absolute poverty households accounted for 25.66% and the incidence of relative poverty was 36.76%. In our sample, the average of the digital finance aggregated index, coverage breadth, and usage depth are 225.30, 203.12, and 246.04, respectively. In terms of control variables, the average age of rural householders is 57.183 and their education level is low, with 13.41% having no formal education. In addition, the household elderly dependency ratio is higher than the child dependency ratio, and only 16.56% of households own a car. Taken together, these are consistent with the basic characteristics of Chinese rural households.

## 4.3. Empirical model

To validate the role of digital finance in rural poverty reduction, we consider the following model:

$$Poverty_{ic} = \beta_0 + \beta_1 DF_c + \beta_2 X_{ic} + \theta_c + \varepsilon_{ic} \tag{1}$$

where the explained variable, $Poverty_{ic}$, indicates whether a rural household $i$ is absolute poverty or relative poverty at the prefecture-level city $c$. The core explanatory variable, $DF_c$, denotes the digital finance indexes of the prefectural-level city $c$. $\beta_1$ is the core coefficient we are concerned with, implying the impact of digital finance on poverty reduction among rural households. $X_{ic}$ refers to control variables that measure householders' characteristics and family characteristics. $\theta_c$ refers to the prefecture-level city fixed effects and $\varepsilon_c$ denotes the error term.

**Table 2. Variable definition and descriptive statistics (N = 11,816).**

| Variables | Definition | Mean | S.D. |
|---|---|---|---|
| Absolute poverty | = 1 if household income per capita is less than 2855 yuan, 0 otherwise | 0.2566 | 0.4368 |
| Relative poverty | = 1 if the household income per capita is in the bottom 25% (including urban households), 0 otherwise | 0.3676 | 0.4822 |
| Digital finance | Digital finance aggregated index at the city level (divided by 100) | 2.2530 | 0.2136 |
| Breadth | Digital financial coverage breadth at the city level (divided by 100) | 2.0312 | 0.2808 |
| Depth | Digital financial usage depth at the city level (divided by 100) | 2.4604 | 0.2497 |
| Age | Age of head of household | 57.183 | 12.096 |
| Age squared | Square of the age of the head of household | 3416.2 | 1394.6 |
| Gender | Female = 1; male = 0 | 0.1119 | 0.3152 |
| Married | = 1 if the head of household is married and has a spouse; 0 otherwise | 0.8731 | 0.3329 |
| Unschooled | Unschooled = 1; schooled = 0 | 0.1341 | 0.3408 |
| Primary school | Primary school = 1; others = 0 | 0.3961 | 0.4891 |
| Junior middle school | Junior middle school = 1; others = 0 | 0.3511 | 0.4773 |
| Senior high school | Senior high school = 1; others = 0 | 0.0897 | 0.2858 |
| Good health | Good health = 1; others = 0 | 0.3758 | 0.4843 |
| Poor health | Poor health = 1; others = 0 | 0.2740 | 0.4460 |
| Ln consumption | Household total consumption, logarithm, yuan | 10.162 | 0.8452 |
| Consumption-income ratio | The ratio of total household consumption divided by income | 3.5499 | 9.9286 |
| Current deposit | Household current deposit, 10,000 yuan | 0.9153 | 2.3891 |
| Fixed deposit | Household fixed deposit, 10,000 yuan | 0.5192 | 2.0836 |
| Debt-income ratio | The ratio of total household debt divided by income | 1.8110 | 6.5349 |
| Child dependency ratio | The number of people over 65 years old as a percentage of the population aged 15–64 in the household | 0.1149 | 0.1872 |
| Elderly dependency ratio | Number of children under 14 years old as a percentage of the population aged 15–64 in the household | 0.2315 | 0.3372 |
| Housing ownership | Owned = 1; not owned = 0 | 0.9336 | 0.2489 |
| Car ownership | Owned = 1; not owned = 0 | 0.1656 | 0.3718 |

It should be noted that by employing a city fixed effect model, we are able to control for endogeneity well, as our explanatory variables are also at the city level. In addition, to control for serial correlation and heterogeneity of variables, we cluster the standard errors to the city level, which also helps to further mitigate the endogeneity problem. Moreover, in the robustness checks, we use an instrumental variable (IV) approach to further rule out potential endogeneity problems.

## 5. Results

### 5.1. The effects of digital finance on rural poverty

We examine the effects of digital finance on rural household poverty in China, and the baseline results are shown in Table 3. Control variables include householders' individual characteristics, household characteristics, and city fixed effects. All standard errors have been clustered at the city level. We employ the regression with two measures of rural household poverty and find that the coefficients on *Digital finance* are both significantly negative in columns (1) and (4), suggesting that digital finance contributes to poverty reduction. Specifically, for each unit increase in the digital finance aggregation index, the probability of absolute household poverty decreases by 10.27% and the probability of relative poverty decreases by 18.31%. Converting the magnitude using standard deviations, the estimates indicate that a one standard deviation increase in digital finance aggregation index reduces absolute poverty by 0.0502 standard deviations and relative poverty by 0.0811 standard deviations.

**Table 3. The impact of digital finance on rural household poverty.**

| | (1) | (2) | (3) | (4) | (5) | (6) |
|---|---|---|---|---|---|---|
| | Absolute poverty | | | Relative poverty | | |
| Digital finance | -0.1027*** | | | -0.1831*** | | |
| | (0.0249) | | | (0.0300) | | |
| Breadth | | -0.0803*** | | | -0.1433*** | |
| | | (0.0195) | | | (0.0235) | |
| Depth | | | -0.1077*** | | | -0.1921*** |
| | | | (0.0261) | | | (0.0315) |
| Age | -0.0115*** | -0.0115*** | -0.0115*** | -0.0146*** | -0.0146*** | -0.0146*** |
| | (0.0029) | (0.0029) | (0.0029) | (0.0031) | (0.0031) | (0.0031) |
| Age squared | 0.0001*** | 0.0001*** | 0.0001*** | 0.0002*** | 0.0002*** | 0.0002*** |
| | (0.0000) | (0.0000) | (0.0000) | (0.0000) | (0.0000) | (0.0000) |
| Gender | 0.0068 | 0.0068 | 0.0068 | -0.0042 | -0.0042 | -0.0042 |
| | (0.0119) | (0.0119) | (0.0119) | (0.0136) | (0.0136) | (0.0136) |
| Married | 0.0346*** | 0.0346*** | 0.0346*** | 0.0296* | 0.0296* | 0.0296* |
| | (0.0126) | (0.0126) | (0.0126) | (0.0154) | (0.0154) | (0.0154) |
| Unschooled | 0.0857*** | 0.0857*** | 0.0857*** | 0.1110*** | 0.1110*** | 0.1110*** |
| | (0.0194) | (0.0194) | (0.0194) | (0.0210) | (0.0210) | (0.0210) |
| Primary school | 0.0498*** | 0.0498*** | 0.0498*** | 0.1037*** | 0.1037*** | 0.1037*** |
| | (0.0149) | (0.0149) | (0.0149) | (0.0178) | (0.0178) | (0.0178) |
| Junior high school | 0.0360** | 0.0360** | 0.0360** | 0.0671*** | 0.0671*** | 0.0671*** |
| | (0.0158) | (0.0158) | (0.0158) | (0.0171) | (0.0171) | (0.0171) |
| Senior high school | 0.0083 | 0.0083 | 0.0083 | 0.0511** | 0.0511** | 0.0511** |
| | (0.0165) | (0.0165) | (0.0165) | (0.0198) | (0.0198) | (0.0198) |
| Good health | -0.0115 | -0.0115 | -0.0115 | -0.0179* | -0.0179* | -0.0179* |
| | (0.0079) | (0.0079) | (0.0079) | (0.0097) | (0.0097) | (0.0097) |
| Poor health | 0.0516*** | 0.0516*** | 0.0516*** | 0.0656*** | 0.0656*** | 0.0656*** |
| | (0.0093) | (0.0093) | (0.0093) | (0.0110) | (0.0110) | (0.0110) |
| Ln consumption | -0.1267*** | -0.1267*** | -0.1267*** | -0.1398*** | -0.1398*** | -0.1398*** |
| | (0.0056) | (0.0056) | (0.0056) | (0.0058) | (0.0058) | (0.0058) |
| Consumption-income ratio | 0.0157*** | 0.0157*** | 0.0157*** | 0.0145*** | 0.0145*** | 0.0145*** |
| | (0.0007) | (0.0007) | (0.0007) | (0.0007) | (0.0007) | (0.0007) |
| Current deposit | -0.0035*** | -0.0035*** | -0.0035*** | -0.0108*** | -0.0108*** | -0.0108*** |
| | (0.0011) | (0.0011) | (0.0011) | (0.0015) | (0.0015) | (0.0015) |
| Fixed deposit | -0.0052*** | -0.0052*** | -0.0052*** | -0.0147*** | -0.0147*** | -0.0147*** |
| | (0.0013) | (0.0013) | (0.0013) | (0.0014) | (0.0014) | (0.0014) |
| Debt-income ratio | 0.0073*** | 0.0073*** | 0.0073*** | 0.0076*** | 0.0076*** | 0.0076*** |
| | (0.0008) | (0.0008) | (0.0008) | (0.0007) | (0.0007) | (0.0007) |
| Child dependency ratio | -0.0263 | -0.0263 | -0.0263 | -0.0104 | -0.0104 | -0.0104 |
| | (0.0204) | (0.0204) | (0.0204) | (0.0246) | (0.0246) | (0.0246) |
| Elderly dependency ratio | -0.0940*** | -0.0940*** | -0.0940*** | -0.0681*** | -0.0681*** | -0.0681*** |
| | (0.0176) | (0.0176) | (0.0176) | (0.0180) | (0.0180) | (0.0180) |
| Housing ownership | -0.0049 | -0.0049 | -0.0049 | -0.0202 | -0.0202 | -0.0202 |
| | (0.0145) | (0.0145) | (0.0145) | (0.0156) | (0.0156) | (0.0156) |
| Car ownership | -0.0163** | -0.0163** | -0.0163** | -0.0690*** | -0.0690*** | -0.0690*** |
| | (0.0078) | (0.0078) | (0.0078) | (0.0120) | (0.0120) | (0.0120) |
| Constant | 1.6547*** | 1.5872*** | 1.6904*** | 3.0680*** | 2.9475*** | 3.1317*** |
| | (0.1119) | (0.1015) | (0.1180) | (0.1305) | (0.1168) | (0.1384) |

(*Continued*)

**Table 3.** (Continued)

|  | (1) | (2) | (3) | (4) | (5) | (6) |
|---|---|---|---|---|---|---|
|  | Absolute poverty | | | Relative poverty | | |
| City fixed effects | Yes | Yes | Yes | Yes | Yes | Yes |
| R-squared | 0.3226 | 0.3226 | 0.3226 | 0.3034 | 0.3034 | 0.3034 |
| N | 11,816 | 11,816 | 11,816 | 11,816 | 11,816 | 11,816 |

Notes: The significance levels of 1%, 5%, and 10% are denoted by ***, **, and *, respectively. Standard errors clustered at the city level are reported in parentheses.

Furthermore, we explore the impact of two sub-indicators of digital finance, breadth coverage and usage depth, on rural household poverty. According to the results in columns (2), (3), (5), and (6), we find that both coverage breadth and usage depth of digital finance can mitigate rural absolute poverty and relative poverty. Specifically, each standard deviation increase in the breadth coverage of digital finance leads to a 0.0516 standard deviation decrease in absolute poverty and a 0.0834 standard deviation decrease in relative poverty for rural households. Similarly, each increase of one standard deviation for the usage depth of digital finance brings about a decrease of 0.0616 standard deviations in absolute poverty and a decrease of 0.0995 standard deviations in relative poverty.

In sum, by using a nationally representative database and city fixed effects models, we confirm the positive impact of digital finance on poverty reduction in China, which are consistent with findings in some previous studies based on other developing country cases [23, 37, 76, 77].

## 5.2. Mechanisms of poverty reduction

**5.2.1. Credit constraints.** As noted above, digital finance provides financial accessibility to rural households and reduces their credit constraints to alleviate poverty. Credit constraint refers to a binary variable indicating whether the household applied for a loan from a bank or credit union, but was rejected. If rural households did experience this situation suggests that they faced credit constraints, the variable is set to 1, and 0 otherwise.

In column (1) of Table 4, the estimates show a significant negative association between digital finance and rural households' credit constraints, implying that an increase in the level of

**Table 4. Digital finance, credit constraints, and rural household poverty.**

|  | (1) | (2) | (3) | (4) | (5) |
|---|---|---|---|---|---|
|  | Credit constraint | | | Absolute poverty | Relative poverty |
| Digital finance | -0.1215*** |  |  |  |  |
|  | (0.0218) |  |  |  |  |
| Breadth |  | -0.0951*** |  |  |  |
|  |  | (0.0170) |  |  |  |
| Depth |  |  | -0.1275*** |  |  |
|  |  |  | (0.0228) |  |  |
| Credit constraint |  |  |  | 0.0358** | 0.0725*** |
|  |  |  |  | (0.0151) | (0.0140) |
| Control variables | Yes | Yes | Yes | Yes | Yes |
| City fixed effects | Yes | Yes | Yes | Yes | Yes |
| N | 11,687 | 11,687 | 11,687 | 11,687 | 11,687 |

Notes: The significance levels of 1%, 5%, and 10% are denoted by ***, **, and *, respectively. Standard errors clustered at the city level are reported in parentheses.
Baseline control variables and city fixed effects are added in all regressions.

**Table 5. Digital financial indicators involving credit and rural household poverty.**

|  | (1) | (2) | (3) | (4) |
|---|---|---|---|---|
|  | Absolute poverty | | Relative poverty | |
| Lending | -0.2417*** | | -0.4312*** | |
|  | (0.0587) | | (0.0707) | |
| Credit | | -0.0528*** | | -0.0941*** |
|  | | (0.0128) | | (0.0154) |
| Control variables | Yes | Yes | Yes | Yes |
| City fixed effects | Yes | Yes | Yes | Yes |
| N | 11,686 | 11,686 | 11,686 | 11,686 |

Notes: The significance levels of 1%, 5%, and 10% are denoted by ***, **, and *, respectively. Standard errors clustered at the city level are reported in parentheses. Baseline control variables and city fixed effects are added in all regressions.

digital finance is effective in alleviating households' credit constraints. In columns (2) and (3), the two digital finance sub-indicators are also negative and significant at the 1% level, indicating that digital financial development reduces the likelihood that rural households experience credit constraint distress. Moreover, in columns (4) and (5), we find that credit constraints are indeed positively associated with household absolute poverty and relative poverty, which is consistent with the findings in previous studies [34, 78–80].

Additionally, as shown in Table 1, digital finance indicator system includes some sub-indexes related to household credit constraints, such as the lending and credit in usage depth; thus, we further consider these indicators to validate the credit constraint mechanism. Table 5 presents the results. We find that both lending and credit reduce poverty among rural households and the estimated coefficients are all significant at the 1% level, suggesting that the credit function of digital finance helps alleviate poverty [28, 33]. All in all, these results provide supportive evidence for Hypothesis 1 and confirm that digital finance could help Chinese rural households escape poverty by easing their credit constraints.

**5.2.2. Information constraints.** As discussed in Section 3, digital finance is based on the Internet and big data technology, which can help rural households alleviate their poverty by alleviating their information constraints. We construct two variables related to household information access, *Information attention* and *Mobile payment*. The former is an ordered variable from 1 to 5, using the householder's concern for economic and financial information, with larger values indicating stronger information concerns. The latter is a binary variable measured by whether rural householders use mobile payment. The reason why mobile payment is regarded as a proxy for information advantages is that mobile payments are becoming an important way for households to access financial and economic information [32, 33].

In the first three columns of Table 6, the estimates suggest that digital finance is positively associated with rural householders' information attentions. Similarly, in the last three columns, the coefficients on *Digital finance* are all positive and statistically significant, indicating that the digital finance similarly increases the probability of mobile payment use by rural households. These results indicate that digital finance increases rural people's attention to economic and financial information, raises their use of mobile payments, and create information advantages for them.

As before, we further test whether these two mechanisms could reduce absolute and relative poverty among rural households, and the results are shown in Table 7. It is clear that all estimated coefficients on *Information attention* and *Mobile payment* are significantly negative, which remains consistent with some literature [81, 82]. These findings provide a preliminary

**Table 6. Information constraints of digital finance.**

| | (1) | (2) | (3) | (4) | (5) | (6) |
|---|---|---|---|---|---|---|
| | Information attention | | | Mobile payment | | |
| Digital finance | 0.4021*** | | | 0.0565** | | |
| | (0.0774) | | | (0.0284) | | |
| Breadth | | 0.3146*** | | | 0.0442** | |
| | | (0.0606) | | | (0.0222) | |
| Depth | | | 0.4219*** | | | 0.0593** |
| | | | (0.0812) | | | (0.0298) |
| Control variables | Yes | Yes | Yes | Yes | Yes | Yes |
| City fixed effects | Yes | Yes | Yes | Yes | Yes | Yes |
| N | 11,786 | 11,786 | 11,786 | 11,816 | 11,816 | 11,816 |

Notes: The significance levels of 1%, 5%, and 10% are denoted by ***, **, and *, respectively. Standard errors clustered at the city level are reported in parentheses. Baseline control variables and city fixed effects are added in all regressions.

indication for the reliability of hypothesis 2, that the information advantage from digital finance helps to alleviate rural household poverty.

Furthermore, since the Internet is the most dominant information exchange platform [54, 83], and digital finance is also used to realize various financial services through the Internet [18, 24, 25], we further introduce a moderator variable, *Internet use*, and construct and interaction term to fully verify the information advantage characteristics of digital finance. In Table 8, the estimates show that although the coefficients on interaction terms are negative in the first three columns, they are insignificant. In contrast, in the last three columns, the coefficients of interaction terms for digital finance and Internet use are all significantly negative, suggesting that digital finance can achieve a reduction in relative poverty among rural households through the information channel of the Internet. Taken together, by using a variety of methods, we support the hypothesis 2 that digital finance is likely to reduce poverty by alleviating information constraints of rural households.

**5.2.3. Social networks.** In Hypothesis 3, we consider that another important mechanism for poverty reduction effect of digital finance is to help expand the social networks of rural households. Given the complexity of social network measurement, several previous studies used money gift income and expenditures and the spending on social network maintenance as proxies for household social networks [50, 84]. The CHFS provides two types of variables in

**Table 7. Information constraints and rural household poverty.**

| | (1) | (2) | (3) | (4) |
|---|---|---|---|---|
| | Absolute poverty | | Relative poverty | |
| Information attention | -0.0370*** | | -0.0247*** | |
| | (0.0033) | | (0.0039) | |
| Mobile payment | | -0.0173** | | -0.0341*** |
| | | (0.0075) | | (0.0122) |
| Control variables | Yes | Yes | Yes | Yes |
| City fixed effects | Yes | Yes | Yes | Yes |
| N | 11,786 | 11,816 | 11,786 | 11,816 |

Notes: The significance levels of 1%, 5%, and 10% are denoted by ***, **, and *, respectively. Standard errors clustered at the city level are reported in parentheses. Baseline control variables and city fixed effects are added in all regressions.

**Table 8. Digital finance, Internet use, and rural household poverty.**

| | (1) | (2) | (3) | (4) | (5) | (6) |
|---|---|---|---|---|---|---|
| | Absolute poverty | | | Relative poverty | | |
| Digital finance | -0.1144*** | | | -0.2264*** | | |
| | (0.0276) | | | (0.0326) | | |
| Digital finance*Internet use | -0.0033 | | | -0.0167*** | | |
| | (0.0034) | | | (0.0047) | | |
| Breadth | | -0.0880*** | | | -0.1754*** | |
| | | (0.0213) | | | (0.0251) | |
| Breadth*Internet use | | -0.0030 | | | -0.0191*** | |
| | | (0.0036) | | | (0.0052) | |
| Depth | | | -0.1203*** | | | -0.2383*** |
| | | | (0.0291) | | | (0.0344) |
| Depth*Internet use | | | -0.0030 | | | -0.0151*** |
| | | | (0.0031) | | | (0.0043) |
| Control variables | Yes | Yes | Yes | Yes | Yes | Yes |
| City fixed effects | Yes | Yes | Yes | Yes | Yes | Yes |
| N | 11,743 | 11,743 | 11,743 | 11,743 | 11,743 | 11,743 |

Notes: The significance levels of 1%, 5%, and 10% are denoted by ***, **, and *, respectively. Standard errors clustered at the city level are reported in parentheses. Baseline control variables and city fixed effects are added in all regressions. *Internet use* is additionally controlled in all columns

terms of income and expenditure associated with social networks. For social network income, we select two variables, *Money gift receive* (dummy) and *Money gift incomes*; for social network expenditure, *Money gift expenditure* (dummy) and *Maintenance expenditure* were selected as mechanism variables. Maintenance expenses related to social networks include transportation expenses, recreation expenses, and communication expenses in 1000 yuan.

Table 9 examines the effects of digital finance on households' social networks from the perspective of income. The estimates in columns (1)-(3) show that there is no association between digital finance and money gift receive of rural households. However, in columns (4)-(6) of

**Table 9. Digital finance, social network, and rural household poverty (revenue related to social networks).**

| | (1) | (2) | (3) | (4) | (5) | (6) | (7) | (8) |
|---|---|---|---|---|---|---|---|---|
| | Money gift receive | | | Money gift incomes | | | Absolute poverty | Relative poverty |
| Digital finance | 0.0084 | | | 5.9673*** | | | | |
| | (0.0374) | | | (0.2265) | | | | |
| Breadth | | 0.0065 | | | 9.7702*** | | | |
| | | (0.0293) | | | (0.3708) | | | |
| Depth | | | 0.0088 | | | 2.8391*** | | |
| | | | (0.0392) | | | (0.1078) | | |
| Money gift incomes | | | | | | | -0.0310*** | -0.0304*** |
| | | | | | | | (0.0040) | (0.0043) |
| Control variables | Yes | Yes | Yes | Yes | Yes | Yes | Yes | Yes |
| City fixed effects | Yes | Yes | Yes | Yes | Yes | Yes | Yes | Yes |
| N | 11,773 | 11,773 | 11,773 | 5236 | 5236 | 5236 | 5236 | 5236 |

Notes: The significance levels of 1%, 5%, and 10% are denoted by ***, **, and *, respectively. Standard errors clustered at the city level are reported in parentheses. Baseline control variables and city fixed effects are added in all regressions.

**Table 10. Digital finance and expenses related to social networks.**

| | (1) | (2) | (3) | (4) | (5) | (6) |
|---|---|---|---|---|---|---|
| | Money gift expenditure | | | Maintenance expenditure | | |
| Digital finance | -1.2337*** | | | 4.3323*** | | |
| | (0.0305) | | | (1.0130) | | |
| Breadth | | -0.9651*** | | | 3.3893*** | |
| | | (0.0239) | | | (0.7925) | |
| Depth | | | -1.2942*** | | | 4.5449*** |
| | | | (0.0320) | | | (1.0627) |
| Control variables | Yes | Yes | Yes | Yes | Yes | Yes |
| City fixed effects | Yes | Yes | Yes | Yes | Yes | Yes |
| N | 11,786 | 11,786 | 11,786 | 11,816 | 11,816 | 11,816 |

Notes: The significance levels of 1%, 5%, and 10% are denoted by ***, **, and *, respectively. Standard errors clustered at the city level are reported in parentheses. Baseline control variables and city fixed effects are added in all regressions.

Table 9, we find that coefficients on *Digital finance* are all positive and significant at the 1% level, implying that the digital finance leads to an increase in money gifts received by rural households. In the last two columns, not surprisingly, the estimates indicate that gift income, as a liquid monetary asset, helps rural households escape poverty.

Moreover, from the social network spending perspective, we further explore whether digital finance can alleviate poverty through social networks. As reported in Table 10, although digital finance significantly reduces the probability of rural households spending on money gifts in columns (1)-(3), it leads to an increase in household spending related to maintaining social networks in the last three columns. Further, in Table 11, we find that money gift expenditure is positively associated with rural household poverty, while there is no association between maintenance expenditure and rural household poverty. These results suggest that while digital finance helps rural households expand their social networks, the additional expenditures incurred may not be conducive to lifting poor rural households out of poverty. Therefore, our findings only partially support Hypothesis 3. However, considering that our measure cannot fully capture all dimensions of social networks of rural households, our estimates provide only suggestive evidence.

**5.2.4. Entrepreneurial activities.** As highlighted in Section 3, another explanation for digital finance to alleviate rural household poverty is entrepreneurial activities. We choose two

**Table 11. Expenses related to social networks and rural household poverty.**

| | (1) | (2) | (3) | (4) |
|---|---|---|---|---|
| | Absolute poverty | | Relative poverty | |
| Money gift expenditure | 0.0520*** | | 0.0537*** | |
| | (0.0090) | | (0.0091) | |
| Maintenance expenditure | | 0.0000 | | 0.0002 |
| | | (0.0002) | | (0.0002) |
| Control variables | Yes | Yes | Yes | Yes |
| City fixed effects | Yes | Yes | Yes | Yes |
| N | 11,786 | 11,816 | 11,786 | 11,816 |

Notes: The significance levels of 1%, 5%, and 10% are denoted by ***, **, and *, respectively. Standard errors clustered at the city level are reported in parentheses. Baseline control variables and city fixed effects are added in all regressions.

**Table 12. Digital finance, entrepreneurship, and rural household poverty.**

| | (1) | (2) | (3) | (4) | (5) | (6) | (7) | (8) |
|---|---|---|---|---|---|---|---|---|
| | Entrepreneurship | | | Online sale | | | Absolute poverty | Relative poverty |
| Digital finance | 0.2062*** | | | 0.0084 | | | | |
| | (0.0318) | | | (0.0101) | | | | |
| Breadth | | 0.1613*** | | | 0.0066 | | | |
| | | (0.0249) | | | (0.0079) | | | |
| Depth | | | 0.2164*** | | | 0.0088 | | |
| | | | (0.0334) | | | (0.0106) | | |
| Entrepreneurship | | | | | | | -0.0212** | -0.0285*** |
| | | | | | | | (0.0085) | (0.0105) |
| Control variables | Yes | Yes | Yes | Yes | Yes | Yes | Yes | Yes |
| City fixed effects | Yes | Yes | Yes | Yes | Yes | Yes | Yes | Yes |
| N | 11,816 | 11,816 | 11,816 | 11,743 | 11,743 | 11,743 | 11,816 | 11,816 |

Notes: The significance levels of 1%, 5%, and 10% are denoted by ***, **, and *, respectively. Standard errors clustered at the city level are reported in parentheses. Baseline control variables and city fixed effects are added in all regressions.

binary variables, namely entrepreneurship and online sale. With the advent of the Internet economy, online sale as a form of informal entrepreneurship has also become popular among Chinese families [33].

In Table 12, we further explore the impact of digital finance on rural households' entrepreneurial activities to test Hypothesis 4. The estimates show that, as expected, digital finance significantly increases rural households' likelihood of entrepreneurship in the first three columns. In addition, the coefficients on *Digital finance* are insignificant in columns (4)-(6), indicating that digital finance does not increase the probability of rural households selling online. Additionally, in columns (7) and (8) of Table 12, the coefficient on *Entrepreneurship* is significantly negative, which indicates that entrepreneurship help rural households to escape from poverty, as emphasized by some previous research [56, 59, 85]. In summary, these estimates support our theoretical expectations in Hypothesis 4 and suggest that digital finance may reduce rural household poverty primarily through offline entrepreneurship.

## 5.3. Heterogeneity analysis

We explore the heterogeneity of digital finance on rural poverty from three perspectives: digital finance usage depth, householder age, and education.

First, in Table 1, the digital financial usage depth includes six sub-items, payment, money funds, lending, insurance, investments, and credit. Among them, lending and credit as mechanisms for digital finance to alleviate rural household credit constraints have been tested, and we further consider the potential heterogeneity effects from payments, money funds, and investments. As shown in Table 13, it is clear that the suppressive effects of these sub-items are significant for both absolute poverty and relative poverty. In comparison, investment and money funds of digital finance are more effective in reducing poverty. The possible reason is that rural households can invest and buy money funds through digital finance without threshold, thus earning higher interest income than bank savings.

Second, given that age is an exogenous variable, introducing an interaction term between digital finance and age does not increase the endogeneity, we construct some interaction models to test the heterogeneity effect of age. The coefficient on the interaction term is significantly negative in column (1) of Table 14, indicating that as the age of householders increases, the

**Table 13. Heterogeneity effects by digital financial usage depth.**

| | (1) | (2) | (3) | (4) | (5) | (6) |
|---|---|---|---|---|---|---|
| | **Absolute poverty** | | | **Relative poverty** | | |
| Payment | -0.0828*** | | | -0.1477*** | | |
| | (0.0201) | | | (0.0242) | | |
| Investment | | -0.1177*** | | | -0.2100*** | |
| | | (0.0286) | | | (0.0344) | |
| Money funds | | | -0.1160*** | | | -0.2069*** |
| | | | (0.0282) | | | (0.0339) |
| Control variables | Yes | Yes | Yes | Yes | Yes | Yes |
| City fixed effects | Yes | Yes | Yes | Yes | Yes | Yes |
| N | 11,816 | 11,816 | 11,816 | 11,816 | 11,816 | 11,816 |

Notes: The significance levels of 1%, 5%, and 10% are denoted by ***, **, and *, respectively. Standard errors clustered at the city level are reported in parentheses. Baseline control variables and city fixed effects are added in all regressions.

effect of digital finance in poverty alleviation is better. Similarly, in column (2), we find that digital financial coverage breadth is also more effective in promoting poverty reduction among older rural households. These results imply that the financial inclusive properties of digital finance in poverty reduction, with positive effects rather better for some socially disadvantaged groups at older ages. However, the last three columns in Table 14 also show that digital finance does not have an age heterogeneity effect in reducing relative poverty, indicating that digital finance may be more helpful to curb the occurrence of absolute poverty for socially vulnerable groups.

Third, we include the interaction term of digital finance and education in the regression, and the results are shown in Table 15. In terms of absolute poverty, the interaction term

**Table 14. Heterogeneity effects by age.**

| | (1) | (2) | (3) | (4) | (5) | (6) |
|---|---|---|---|---|---|---|
| | **Absolute poverty** | | | **Relative poverty** | | |
| Digital finance | 0.0654 | | | -0.1862 | | |
| | (0.0952) | | | (0.1155) | | |
| Digital finance*Age | -0.0035* | | | 0.0001 | | |
| | (0.0018) | | | (0.0022) | | |
| Breadth | | 0.0435 | | | -0.1474* | |
| | | (0.0710) | | | (0.0861) | |
| Breadth*Age | | -0.0026* | | | 0.0001 | |
| | | (0.0014) | | | (0.0017) | |
| Depth | | | -0.0064 | | | -0.2181** |
| | | | (0.0840) | | | (0.0975) |
| Depth*Age | | | -0.0021 | | | 0.0005 |
| | | | (0.0016) | | | (0.0018) |
| Control variables | Yes | Yes | Yes | Yes | Yes | Yes |
| City fixed effects | Yes | Yes | Yes | Yes | Yes | Yes |
| N | 11,816 | 11,816 | 11,816 | 11,816 | 11,816 | 11,816 |

Notes: The significance levels of 1%, 5%, and 10% are denoted by ***, **, and *, respectively. Standard errors clustered at the city level are reported in parentheses. Baseline control variables and city fixed effects are added in all regressions.

**Table 15. Heterogeneity effects by education.**

| | (1) | (2) | (3) | (4) | (5) | (6) |
|---|---|---|---|---|---|---|
| | Absolute poverty | | | Relative poverty | | |
| Digital finance | -0.1028*** | | | -0.1830*** | | |
| | (0.0248) | | | (0.0300) | | |
| Digital finance*Unschooled | -0.0932* | | | 0.0639 | | |
| | (0.0500) | | | (0.0447) | | |
| Breadth | | -0.0803*** | | | -0.1433*** | |
| | | (0.0194) | | | (0.0235) | |
| Breadth*Unschooled | | -0.0834** | | | 0.0381 | |
| | | (0.0404) | | | (0.0370) | |
| Depth | | | -0.1080*** | | | -0.1917*** |
| | | | (0.0261) | | | (0.0315) |
| Depth*Unschooled | | | -0.0525 | | | 0.0656* |
| | | | (0.0409) | | | (0.0370) |
| Control variables | Yes | Yes | Yes | Yes | Yes | Yes |
| City fixed effects | Yes | Yes | Yes | Yes | Yes | Yes |
| N | 11,816 | 11,816 | 11,816 | 11,816 | 11,816 | 11,816 |

Notes: The significance levels of 1%, 5%, and 10% are denoted by ***, **, and *, respectively. Standard errors clustered at the city level are reported in parentheses. Baseline control variables and city fixed effects are added in all regressions.

between digital finance and uneducated is significantly negative in column (1), suggesting that digital finance is more conducive to uneducated rural households to escape poverty. Similarly, interaction terms are significantly negative in column (2) and insignificant in column (3), indicating that increasing the coverage breadth of digital finance may benefit more socially disadvantaged groups. In the last three columns, the three interaction terms remain insignificant, implying that for relative poverty, the financial inclusive properties of digital finance are not fully exploited.

## 5.4. Robustness checks

**5.4.1. IV methods.** Although we control for city fixed effects and cluster at the city level, some potential endogeneity problems could not be completely ruled out. Therefore, we adopt the IV methods to perform robustness tests. Referring to previous studies [25, 35], we use provincial Internet penetration as an IV, and the original data were obtained from the Statistical Report on the Internet Development in China.

A good instrumental variable needs to satisfy both relevance assumption and exclusion restriction assumption. From the perspective of relevance assumption, the diffusion and popularity of the Internet is an important basic condition for the development of digital finance [28, 35], and digital finance tends to grow better in regions with better Internet infrastructure in China [18, 27]. Therefore, Internet penetration and digital finance development are closely linked. In terms of the exclusion restriction hypothesis, considering that some previous studies concluded the role of Internet infrastructure in poverty alleviation [81, 82, 86], we use historical Internet penetration as an IV. Since the earliest data provided by the Statistical Report on Internet Development in China is 1997, we use the provincial Internet penetration in 1997 as the IV. After controlling for the city fixed effects, it is difficult for historical provincial Internet penetration to directly affect household poverty through other channels, which makes our selected IV theoretically feasible.

**Table 16. The impact of digital finance on rural household poverty: IV methods (first-stage results).**

|  | (1) | (2) | (3) |
|---|---|---|---|
|  | Digital finance | Breadth | Depth |
| Historical Internet penetration | 0.3014*** | 0.3620*** | 0.3650*** |
|  | (0.0832) | (0.0843) | (0.1315) |
| Control variables | Yes | Yes | Yes |
| City fixed effects | Yes | Yes | Yes |
| First-stage F value | 13.1176 | 18.4637 | 7.7104 |
| N | 11,816 | 11,816 | 11,816 |

Notes: The significance levels of 1%, 5%, and 10% are denoted by ***, **, and *, respectively. Standard errors clustered at the city level are reported in parentheses. Baseline control variables and city fixed effects are added in all regressions.

We employ the two stage least square (2SLS) method, and the results of the first stage are shown in Table 16. We find the IV, historical Internet penetration, is positively correlated with Digital finance, with statistical significance at the 1% level. More importantly, the first-stage F value in the first two columns is well above the Stock-Yogo critical value for a weak IV [87], and in column (3), the first-stage F value less than 10. In summary, the first-stage estimated results indicate that historical Internet penetration contributes to the digital finance development in China.

Table 17 shows the second stage results. Not surprisingly, in the second-stage results, the Anderson-Rubin Wald test suggests that our IV is strong (the P-value is less than 0.05), and all the coefficients of the variables related to digital finance are significantly negative at the 1% level. Based on columns (1) and (4), the IV estimates suggest that for each unit increase in the digital finance aggregation index, the probability of absolute poverty and relative poverty among rural households decreases by 9.50% and 16.84%, respectively, which is quite close to the OLS estimates in Table 3. Thus, the IV estimates suggest that our main specification is robust and digital finance does play an important role in reducing poverty in rural China.

**5.4.2. Using alternative specifications.** First, as discussed in Section 4, the current definition of relative poverty is not uniform. It may be too arbitrary for us to consider the bottom

**Table 17. The impact of digital finance on rural household poverty: IV methods (second-stage results).**

|  | (1) | (2) | (3) | (4) | (5) | (6) |
|---|---|---|---|---|---|---|
|  | Absolute poverty |  |  | Relative poverty |  |  |
| Digital finance | -0.0950*** |  |  | -0.1684*** |  |  |
|  | (0.0239) |  |  | (0.0287) |  |  |
| Breadth |  | -0.0744*** |  |  | -0.1318*** |  |
|  |  | (0.0187) |  |  | (0.0225) |  |
| Depth |  |  | -0.0997*** |  |  | -0.1767*** |
|  |  |  | (0.0250) |  |  | (0.0302) |
| Control variables | Yes | Yes | Yes | Yes | Yes | Yes |
| City fixed effects | Yes | Yes | Yes | Yes | Yes | Yes |
| Anderson-Rubin Wald test | 4.2934 | 4.2934 | 4.2934 | 42.0415 | 42.0415 | 42.0415 |
| P-value | 0.0383 | 0.0383 | 0.0383 | 0.0000 | 0.0000 | 0.0000 |
| N | 11,816 | 11,816 | 11,816 | 11,816 | 11,816 | 11,816 |

Notes: The significance levels of 1%, 5%, and 10% are denoted by ***, **, and *, respectively. Standard errors clustered at the city level are reported in parentheses. Baseline control variables and city fixed effects are added in all regressions.

**Table 18. Robustness checks by redefining relative poverty.**

|  | (1) | (2) | (3) | (4) | (5) | (6) |
|---|---|---|---|---|---|---|
| Digital finance | -0.1453*** |  |  | -0.2709*** |  |  |
|  | (0.0264) |  |  | (0.0347) |  |  |
| Breadth |  | -0.1137*** |  |  | -0.2119*** |  |
|  |  | (0.0207) |  |  | (0.0272) |  |
| Depth |  |  | -0.1524*** |  |  | -0.2842*** |
|  |  |  | (0.0277) |  |  | (0.0364) |
| Control variables | Yes | Yes | Yes | Yes | Yes | Yes |
| City fixed effects | Yes | Yes | Yes | Yes | Yes | Yes |
| N | 11,816 | 11,816 | 11,816 | 11,816 | 11,816 | 11,816 |

Notes: The significance levels of 1%, 5%, and 10% are denoted by ***, **, and *, respectively. Standard errors clustered at the city level are reported in parentheses. Baseline control variables and city fixed effects are added in all regressions.

25% of household income per capita as relative poverty. Therefore, in the robustness tests, we redefine the bottom 15% and 35% as relative poverty, respectively. In Table 18, we find that the empirical results remain unchanged.

Second, we adopt another database, the Chinese Thousand Village Survey (CTVS) data from to conduct another robustness test. Similar to the CHFS, the CTVS revolves around the socioeconomic status of rural households, providing not only a large number of demographic characteristics of individuals, but also including a large amount of information on household economics [88]. By matching digital financial indexes of the prefectural-level cities, we can similarly explore the impact of digital finance on rural poverty. Since the CTVS does not provide specific values for household debt and household savings, we remove the control variable *Debt-income ratio* and replace *Current deposit* and *Fixed deposit* with dummy variables. As shown in column (1) of Table 19, the coefficients on *Digital finance* are all significantly negative in two Panels, indicating that our main results are quite robust for using an alternative database.

Third, in rural China, the elderly is more likely to fall into poverty because of the greater health shocks and income risks [1, 89]. In other words, age may become one of the core factors affecting poverty in rural Chinese families. Therefore, we try to further control the age fixation effect. In column (2) of Table 19, we find that our main results are satisfactory for capturing age fixed effects.

Fourth, considering that the household poverty in rural China may differ greatly by health status and between regions, we try to include the province-by-health fixed effects for a robustness check [90, 91]. In column (3) of Table 19, we find that this change has little effect on the results, suggesting that our main results are convincing by controlling a high-dimensional fixed effect.

Lastly, given that digital finance, as an emerging financial tool, may have spillover effects and peer group effect in small areas [92]. Therefore, we consider adding some village-level control variables, including the average age, average education, and average health status of householders, as well as the average family savings (including current and fixed deposits). In column (4) of Table 19, the estimates are qualitatively similar to the main estimates. In addition, we use the village level clustering standard errors for a further robustness check. In column (5) of Table 19 we find that the three indicators of digital finance remain significantly negative at the 1% level in both two Panels.

**Table 19. Robustness checks by excluding extreme observations or using alternative specification.**

|  | (1) | (2) | (3) | (4) | (5) |
|---|---|---|---|---|---|
|  | Using an alternative database | Controlling for age fixed effects | Controlling for province-by-health fixed effects | Adding some village control variables | Clustering at the village level |
| **Panel A. Absolute poverty** |  |  |  |  |  |
| Digital finance | -0.1022** | -0.1633*** | -0.1790*** | -0.0888*** | -0.1390*** |
|  | (0.0498) | (0.0510) | (0.0292) | (0.0316) | (0.0300) |
| Breadth | -0.0833* | -0.1278*** | -0.1400*** | -0.0694*** | -0.1087*** |
|  | (0.0463) | (0.0399) | (0.0229) | (0.0247) | (0.0234) |
| Depth | -0.1223** | -0.1714*** | -0.1878*** | -0.0931*** | -0.1458*** |
|  | (0.0631) | (0.0535) | (0.0307) | (0.0331) | (0.0314) |
| **Panel B. Relative poverty** |  |  |  |  |  |
| Digital finance | -0.1877** | -0.2344*** | -0.4428*** | -0.1472*** | -0.2166*** |
|  | (0.1023) | (0.0591) | (0.0357) | (0.0353) | (0.0327) |
| Breadth | -0.1997** | -0.1834*** | -0.3464*** | -0.1152*** | -0.1695*** |
|  | (0.0942) | (0.0462) | (0.0279) | (0.0276) | (0.0256) |
| Depth | -0.2765* | -0.2459*** | -0.4645*** | -0.1544*** | -0.2273*** |
|  | (0.1644) | (0.0620) | (0.0375) | (0.0370) | (0.0343) |

Notes: The significance levels of 1%, 5%, and 10% are denoted by ***, **, and *, respectively. Standard errors clustered at the city level are reported in parentheses except column (4). Baseline control variables and city fixed effects are added in all regressions.

# 6. Conclusion and discussion

The poverty problem in rural China has long been a concern for the government and the community. Digital finance, as a new financial format that can reach more socially vulnerable groups, may become a new direction to reduce poverty in rural China. By matching digital financial indexes of the prefectural-level cities and rural household microdata from the CHFS in 2017, we examine the role of digital finance in alleviating rural household poverty using a city fixed effect approach.

The results indicate that digital financial significantly reduce absolute poverty and relative poverty among Chinese rural households, which is supported by a series of robustness tests. Specifically, our estimates show that for each unit increase in the digital finance aggregation index, the probability of absolute and relative poverty in rural households decreases by 10.27% and 18.31%, respectively. Mechanism analysis results show that digital finance alleviates credit constraints and information constraints of rural households, widens their social networks, and promotes entrepreneurship, which further help them to curb poverty problems. Moreover, we find that the development of payments, investment, and money funds in digital finance all contribute to rural households' poverty reductions, but for elderly and uneducated socially disadvantaged groups, the role of digital finance is limited to mitigating absolute poverty.

The relevant policy implications are as follows. First, our results indicate that digital finance has a significant effect on the alleviation of relative poverty. Therefore, the government should further promote the construction of digital financial infrastructure in underdeveloped regions through government financial support and guidance of the related policy, such as increasing smartphone penetration, accelerating the construction of 5G networks and the application of big data technologies, and enable digital finance to benefit more low-income and poor groups. Second, our findings suggest that digital finance does not appear to be sufficient in alleviating the relative poverty of some older and uneducated people. The government's poverty

alleviation department proposes to establish some cooperative projects with research institutions and digital financial institutions to investigate the difficulties and needs of the elderly and low-educated people in using digital financial services, and further improve the platform, which is more beneficial to disadvantaged groups.

Additionally, our findings not only serve China, but are also instructive for other developing countries. Evidence from China suggests that the inclusive nature of digital finance can reach more poor people and may help alleviate the financing constraints and information constraints and promote their entrepreneurial activities of rural poverty. Therefore, this paper may be supportive of increased investment in digital finance to alleviate poverty in some developing countries and low-income countries, and provide a new direction for related public policies.

However, there are some limitations in this paper. First, since the digital finance index is compiled considering the entire administrative area of cities, it does not distinguish between urban and rural areas. Therefore, compared with the real status of digital financial development in rural China, the indicators we use may be on the high side. With the improvement and refinement of digital financial indicators, this problem is expected to be improved in future studies. Second, the mechanism variables may not be comprehensive and perfect. For example, in the measurement of social networks of rural households, our analysis only from the perspective of gift money may not be sufficient. Subsequent research might be supplemented by network lending and social interaction. Third, since the latest data available from CHFS is 2017, we are unable to use more recent data. Follow-up literature can update the data to further complement our study.

## Acknowledgments

The authors wish to thank the editors and two anonymous referees for comments that considerably improved the quality of this paper. The authors acknowledge the data support from the China Household Finance Survey (CHFS) carried out by the Survey and Research Center for China Household Finance in Southwest University of Finance and Economics, the digital financial indexes from the Institute of Digital Finance of Peking University, and the Chinese Thousand Village Survey (CTVS) of Shanghai University of Finance and Economics.

## Author Contributions

**Conceptualization:** Chunkai Zhao.

**Data curation:** Boou Chen, Chunkai Zhao.

**Formal analysis:** Chunkai Zhao.

**Funding acquisition:** Boou Chen, Chunkai Zhao.

**Methodology:** Boou Chen, Chunkai Zhao.

**Software:** Boou Chen, Chunkai Zhao.

**Supervision:** Boou Chen.

**Visualization:** Boou Chen.

**Writing – original draft:** Chunkai Zhao.

**Writing – review & editing:** Boou Chen.

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
