## [Decision Letter · Decision Letter 0]

27 Aug 2021

PONE-D-21-15370

Poverty Reduction in Rural China: Does the Digital Finance Matter?

PLOS ONE

Dear Dr. Chen,

Thank you for submitting your manuscript to PLOS ONE. After careful consideration, we feel that it has merit but does not fully meet PLOS ONE’s publication criteria as it currently stands. Therefore, we invite you to submit a revised version of the manuscript that addresses the points raised during the review process.

We look forward to receiving your revised manuscript.

Kind regards,

Mingxing Chen, Ph.D.

Academic Editor

PLOS ONE

Journal Requirements:

Reviewers' comments:

Reviewer's Responses to Questions

**Comments to the Author**

1. Is the manuscript technically sound, and do the data support the conclusions?

Reviewer #1: Yes

Reviewer #2: Partly

2. Has the statistical analysis been performed appropriately and rigorously? 

Reviewer #1: Yes

Reviewer #2: Yes

3. Have the authors made all data underlying the findings in their manuscript fully available?

Reviewer #1: Yes

Reviewer #2: Yes

4. Is the manuscript presented in an intelligible fashion and written in standard English?

Reviewer #1: Yes

Reviewer #2: No

5. Review Comments to the Author

Reviewer #1: I am positively disposed to the fundamental ideas of the paper. My suggestions are as follows:

1. In Introduction, the relevant research on China targeted measures in poverty alleviation should be detailed.

2. Information advantages is brought by digital finance? Or caused by Information and communication technology? The authors should provide more evidence for Hypothesis 2 and explain more about the effects of digital finance on financial information advantages. Similarly, the other hypothesizes should be considered from the perspective. A diagram where the relationship between digital finance and poverty reduction are clearly described is needed. The credit constraints, information advantage, social networks, and entrepreneurship should be placed in the diagram.

3. If the data，such as the 2016 digital finance aggregation index, could be updated to the recent years, the research would be better.

4. There is a minor mistake in row 328, “and” is redundancy.

5. The distance from each city to Hangzhou was chosen as an instrumental variable. Geography distance is not the most important, but time distance and information distance may be more meaningful.

Reviewer #2: 1. To better clarify the significance of this research, the logic of the introduction needs to be strengthened, and the review of existing research should focus more on the research theme.

2. The analysis of the theoretical framework is unconvincing. For example, can digital finance bring information advantages to the poor? In fact, the poor do not have information advantages, and digital finance only reduces information inequality in a sense.

3. The spatial scale of DFII data is province, but the empirical analysis takes the prefecture-level city as the spatial unit. Therefore, this is questionable.

4. The empirical analysis of antipoverty mechanism is not deep and persuasive.

5. “5.4 Robustness checks”: the logarithm of the distance to Hangzhou? This analysis is not credible. Please reconsider relevant content.

6. In the past, a large number of the poor in rural China were old, weak, sick and disabled, but they were excluded in this study. This makes the results questionable.

7. The policy implications is not targeted and needs to be strengthened. For example, digital financial infrastructure is seriously insufficient in less developed countries, and their first problem is to promote the construction of digital financial infrastructure. However, the research only outlines the need to strengthen digital finance, but did not analyze how to achieve it. Therefore, the policy enlightenment is unrealistic.

6. PLOS authors have the option to publish the peer review history of their article (what does this mean?). If published, this will include your full peer review and any attached files.

Reviewer #1: No

Reviewer #2: No

---

## [Author Response · Author response to Decision Letter 0]

4 Oct 2021

“Poverty Reduction in Rural China: Does the Digital Finance Matter?”

Response to Reviewer #1

We appreciate that our reviewer provides meaningful suggestions and comments on several details of our study, which further helps us to improve our paper. In this revised version, we attempt to address all the concerns our reviewer proposes. Our point-by-point responses to the reviewer are as follows. 

The points raised by reviewers are written in blue italics, whereas our responses are shown in normal font (single-spaced), and the key quotation of the revised manuscript is shown in red font (double-spaced). In the modified manuscript, all changes are marked in red.

1. In Introduction, the relevant research on China targeted measures in poverty alleviation should be detailed.

Response:

Based on the reviewer’s comments, in the revised version, we revised and improved the section 1, Introduction. In the first paragraph, it specifically stated the history of China’s poverty reduction and related literature, especially highlighting relevant research on targeted poverty alleviation in details.

The revised content is as follows (on pages 2-3):

…

Poverty reduction is the basis for maintaining social stability and has become one of the major challenges in developing countries. China is the largest developing country in the world and once had the largest rural poor population (Liu et al., 2017). Since 1949, China has made great efforts to solve the problems of poverty and implemented a series of poverty reduction measures in different stages. Before 1978, the primary objective of antipoverty was to ensure basic survival of farmers, and the main measures were low-level social assistance together with mutual aid and cooperation (Guo et al., 2019). However, in 1978, according to the rural poverty standard calculated at the price level of that year, 770 million people are still in absolute poverty, accounting for 97.5% of the rural population. From 1978 to 2012, China's institutional reform had significantly relieved the poverty in rural areas, more than 700 million people in rural China overcame the problems of poverty. In 2013, the Chinese government implemented the targeted poverty alleviation (TPA). The TPA ensured that assistance accurately reaches poverty-stricken villages and households, and combined five approaches to eliminate poverty, which are industrial development, resettlement, ecological compensation, strengthened education and social security (Guo et al., 2019; Liao et al., 2021; Liu et al., 2018; Zhou et al., 2018). The latest report from the China's National Bureau of Statistics shows that from 2012 to 2019, the average annual reduction rate of rural poverty was as high as 51.06%, and problem of absolute poverty was completely solved in 2020. However, the relative poverty of rural households remains severe due to the large disparity between urban and rural development in China (Peng et al., 2021; Wang et al., 2020). 

…

2. Information advantages is brought by digital finance? Or caused by Information and communication technology? The authors should provide more evidence for Hypothesis 2 and explain more about the effects of digital finance on financial information advantages. Similarly, the other hypothesizes should be considered from the perspective. A diagram where the relationship between digital finance and poverty reduction are clearly described is needed. The credit constraints, information advantage, social networks, and entrepreneurship should be placed in the diagram.

Response:

We sincerely appreciate our reviewer’s suggestion. Follow the reviewer’s suggestion, we summarized and drew a diagram of the influence mechanism (Fig 3), and showed it in the last paragraph of the Section 3 “Theoretical framework”. The diagram is shown below:

Fig 3. The impact mechanism of digital finance on poverty reduction

In addition, we have carefully revised the section 3.2, changed the title from “Information advantages” to “Information constraints”, and the literature has been added to support the Hypothesis 2. In the revised version, in response to reviewers’ suggestions, we emphasized that digital finance is a new financial format that combines the ICT with traditional financial services. Different from the pure impact of ICT, with the help of financial platforms and big data technology, digital finance can deliver information that is more useful to clients to improve their economic conditions and is more compatible with user characteristics, further alleviating the information constraints of poor people.

The revised content is as follows (on page 7):

…

3.2. Information constraints

In addition to credit constraints, poor and low-income rural households also face strong information constraints. There is a clear “digital gap” with middle-income and high-income groups in the production, employment, and life for the poor. Some studies confirmed the impact of the digital divide on the income gap (Chinn and Fairlie, 2010; Kiiski and Pohjola, 2002; Quibria et al., 2003). On the contrary, with the rapid development of information and communication technology (ICT), the use of smartphones and the Internet has a significant role in increasing individual income (Krueger, 1993; DiMaggio and Bonikowski, 2008). Digital finance is a new financial format that combines the ICT with traditional financial services to reach more groups (Lai et al., 2020). Therefore, the development of digital finance may further strengthen the role of ICT in narrowing the income gap and further promote poverty reduction by alleviating the information constraints of poor and low-income households.

Furthermore, low-income people usually lack financial knowledge and have limited ability to collect and identify data from the Internet. Therefore, although the development of ICT makes it easier to obtain information and reduces the cost of obtaining information, it may still be difficult to benefit low-income groups. With the help of financial platforms and big data technology, digital finance will deliver information that is more useful to clients to improve their economic conditions and is more compatible with user characteristics (Guo et al., 2020; Xie et al., 2018). People can easily obtain information related to agricultural production and management, employment, finance and daily life timely from digital financial platforms (Wang, 2020; Yin et al., 2019). After big data analysis, this part of information is highly matched with users, more accurate and transparent. It may help to promote the employment of rural laborers and improve the efficiency of agricultural production (Liu et al., 2021), thus increase their income and reducing the incidence of poverty. In addition, even if the information received is only about daily life, rural households have the opportunity to reallocate resources optimally and improve their ability to cope with external risk shocks (Huang and Huang, 2018). To sum up, we propose the second hypothesis: 

Hypothesis 2: Digital finance is likely to curb rural poverty by leveraging information and alleviating information constraints.

…

Furthermore, we have also carefully revised the statement of other hypothesizes, added more literature, and provided more evidence for them. The revised content is as follows (on pages 6-9):

…

3.1. Credit constraints

Digital finance may reduce the incidence of poverty by alleviating credit constraints. Low-income and poor rural households often have strong credit constraints and are affected by lack of access to the inadequate provision of financial services, making it difficult to improve their economic conditions (Imai et al., 2010). Traditional financial institutions have high unit costs for granting agricultural credit and lower overall returns (Berger and Udell, 2002), while rural households live more dispersedly, and loans from rural households and micro enterprises are often in a small scale. Therefore, poor rural households are difficult to achieve the formal financial services from traditional financial institutions, and unable to obtain additional and funds for production or other investments (Shoji et al., 2012).

Compared to traditional financial institutions, digital finance only need less investment for system construction and development at the initial stage, and can reduce the degree of information asymmetry and the risk of adverse selection by integrating a large number of online user information (Beck et al., 2018). It further promotes the development of financial inclusion, and reduce the rate of financial exclusion among the poor. In addition, benefit from digital finance, loan application only needs to be completed on the Internet terminals, which is more convenient and friendly for the rural households with limited financial knowledge (Lai et al., 2020). Digital finance helps poor rural households alleviate their credit constraints by increasing their possibilities of achieving financial services and simplifying the process of loan application. The alleviation of credit constraints on rural households may increase the family income and improve their ability to bear risks, which reduce the incidence of poverty (Jack et al., 2013). Therefore, we put forward the first hypothesis:

Hypothesis 1: Digital finance may reduce rural poverty by alleviating credit constraints.

…

3.3. Social networks

Digital finance may help rural households expand social networks and strengthen ties with relatives and friends. In China, social networks are important institutional social capital that could explain the role of digital financial development in alleviating rural household poverty. Previous literature suggested that social networks were closely related to individuals' income, employment, and occupation choices (Montgomery, 1991; Zhang and Li, 2003). In a typical relational society, social networks even play an important role in lifting rural Chinese families out of poverty (Klärner and Knabe, 2019; Zhang et al., 2017).

The digital finance has provided people with a more convenient way to pay and increased the frequency of social engagement. Relying on the Internet platform, digital finance provides people with an effective means of communication and social interaction. For example, WeChat Pay was developed by relying on WeChat, the largest online social platform in China. By combining the custom of WeChat red envelopes with traditional Chinese features, it has greatly enhanced the online social interaction experience (Matemba et al., 2018). Additionally, digital financial development has the potential to increase people's online accessibility and facilitate their participation in online social networking (Hsiao, 2011; Liébana-Cabanillas et al., 2018). Thus we derive the third hypothesis:

Hypothesis 3: Digital finance is likely to reduce rural poverty by expanding social networks.

3.4. Entrepreneurial activities

Digital finance may alleviate poverty by promoting entrepreneurial activities of rural households. Entrepreneurial activities as a solution to reduce poverty has been explored by many research (e.g., Bruton et al., 2013; He, 2019; Si et al., 2015; Sutter et al., 2019). Entrepreneurship, especially informal entrepreneurship, as an important source of increasing household income in China, is an effective way to get rural households out of the poverty trap (He, 2019; Si et al., 2015). However, strong credit constraints will hinder entrepreneurial behavior, especially for low-income and poor families (e.g., Corradin and Popov, 2015; Evans and Jovanovic, 1989; Karaivanov, 2012). The financing function of digital finance improves the credit availability of potential entrepreneurs (Bianchi, 2010), and has a positive impact on rural households' entrepreneurial activities (Wang, 2020). With the help of digital financial platforms, entrepreneurial farmers can obtain a large amount of information related to entrepreneurship, and strengthen cooperation with buyers or other entrepreneurs, so as to evaluate accurately the feasibility and market prospects of entrepreneurial projects (Xie et al., 2018). In addition, mobile payment can reduce transaction costs and make transactions more convenient and safer (Jack and Suri, 2014; Suri, 2017). The reduction of transaction costs and transaction risks increases the potential returns of entrepreneurs (Beck et al., 2018). In summary, we formulate the fourth hypothesis: 

Hypothesis 4: Digital finance may alleviate poverty by promoting entrepreneurial activities of rural households.

…

3. If the data，such as the 2016 digital finance aggregation index, could be updated to the recent years, the research would be better.

Response:

 We quite agree with the reviewer's comment that if we could updated the data to the recent years, the research would be better. As shown in Figs 1 and 2, the latest data of digital finance aggregation index has been updated to 2018. However, the latest publicly available data for CHFS is 2017, so we had to use this 2017 data in our empirical analysis. In addition, to reduce endogeneity, we used the macro data of digital finance aggregation index in 2016. In Section 6, we pointed out some shortcomings of our study and provided some directions for future research, including the issue of data updating. 

The revised content is as follows (on page 30):

…

However, there are some limitations in this paper. First, since the digital finance index is compiled considering the entire administrative area of cities, it does not distinguish between urban and rural areas. Therefore, compared with the real status of digital financial development in rural China, the indicators we use may be on the high side. With the improvement and refinement of digital financial indicators, this problem is expected to be improved in future studies. Second, the mechanism variables may not be comprehensive and perfect. For example, in the measurement of social networks of rural households, our analysis only from the perspective of gift money may not be sufficient. Subsequent research might be supplemented by network lending and social interaction. Third, since the latest data available from CHFS is 2017, we are unable to use more recent data. Follow-up literature can update the data to further complement our study.

…

4. There is a minor mistake in row 328, “and” is redundancy.

Response:

 Many thanks to the reviewer for your careful reading, and this writing error has been corrected in the revised manuscript.

5.The distance from each city to Hangzhou was chosen as an instrumental variable. Geography distance is not the most important, but time distance and information distance may be more meaningful.

Response:

We sincerely appreciate our reviewer’s suggestion. Follow the reviewer’s suggestion, we replaced the instrumental variable (IV) in the revised manuscript. Referring to previous studies (Li et al., 2020; Xie et al., 2018), we use provincial historical Internet penetration as an IV. 

The revised content is as follows (on pages 25-27):

…

5.4.1. IV methods

Although we control for city fixed effects and cluster at the city level, some potential endogeneity problems could not be completely ruled out. Therefore, we adopt the IV methods to perform robustness tests. Referring to previous studies (Li et al., 2020; Xie et al., 2018), we use provincial Internet penetration as an IV, and the original data were obtained from the Statistical Report on the Internet Development in China. 

A good instrumental variable needs to satisfy both relevance assumption and exclusion restriction assumption. From the perspective of relevance assumption, the diffusion and popularity of the Internet is an important basic condition for the development of digital finance (Liu et al., 2021; Xie et al., 2020), and digital finance tends to grow better in regions with better Internet infrastructure in China (Guo et al., 2020; Huang and Tao, 2019). Therefore, Internet penetration and digital finance development are closely linked. In terms of the exclusion restriction hypothesis, considering that some previous studies concluded the role of Internet infrastructure in poverty alleviation (e.g., Chao et al., 2021; Galperin and Viecens, 2017; James, 2006; Mora-Rivera and García-Mora, 2021), we use historical Internet penetration as an IV[ Since the earliest data provided by the Statistical Report on Internet Development in China is 1997, we use the provincial Internet penetration in 1997 as the IV. ]. After controlling for the city fixed effects, it is difficult for historical provincial Internet penetration to directly affect household poverty through other channels, which makes our selected IV theoretically feasible.

We employ the two stage least square (2SLS) method, and the results of the first stage are shown in Table 16. We find the IV, historical Internet penetration, is positively correlated with Digital finance, with statistical significance at the 1% level. More importantly, the first-stage F value in the first two columns is well above the Stock-Yogo critical value for a weak IV (Stock and Yogo, 2005)[ In column (3), the first-stage F value less than 10. In the second-stage results, the Anderson-Rubin Wald test suggests that our IV is strong (the P-value is less than 0.05).]. In summary, the first-stage estimated results indicate that historical Internet penetration contributes to the digital finance development in China.

Table 16. The impact of digital finance on rural household poverty: IV methods (first-stage results)

 (1) (2) (3)

 Digital finance Breadth Depth

Historical Internet penetration 0.3014*** 0.3620*** 0.3650***

 (0.0832) (0.0843) (0.1315)

Control variables Yes Yes Yes

City fixed effects Yes Yes Yes

First-stage F value 13.1176 18.4637 7.7104

N 11,816 11,816 11,816

Notes: The significance levels of 1%, 5%, and 10% are denoted by ***, **, and *, respectively. Standard errors clustered at the city level are reported in parentheses. Baseline control variables and city fixed effects are added in all regressions.

Table 17 shows the second stage results. Not surprisingly, all the coefficients of the variables related to digital finance are significantly negative at the 1%level. Based on columns (1) and (4), the IV estimates suggest that for each unit increase in the digital finance aggregation index, the probability of absolute poverty and relative poverty among rural households decreases by 9.5% and 16.84%, respectively, which is quite close to the OLS estimates in Table 3. Thus, the IV estimates suggest that our main specification is robust and digital finance does play an important role in reducing poverty in rural China.

Table 17. The impact of digital finance on rural household poverty: IV methods (second-stage results)

 (1) (2) (3) (4) (5) (6)

 Absolute poverty Relative poverty

Digital finance -0.0950*** -0.1684*** 

 (0.0239) (0.0287) 

Breadth -0.0744*** -0.1318*** 

 (0.0187) (0.0225) 

Depth -0.0997*** -0.1767***

 (0.0250) (0.0302)

Control variables Yes Yes Yes Yes Yes Yes

City fixed effects Yes Yes Yes Yes Yes Yes

Anderson-Rubin Wald test 4.2934 4.2934 4.2934 42.0415 42.0415 42.0415

P-value 0.0383 0.0383 0.0383 0.0000 0.0000 0.0000

N 11,816 11,816 11,816 11,816 11,816 11,816

Notes: The significance levels of 1%, 5%, and 10% are denoted by ***, **, and *, respectively. Standard errors clustered at the city level are reported in parentheses. Baseline control variables and city fixed effects are added in all regressions.

…

References

1.Beck, T., Pamuk, H., Ramrattan, R., & Uras, B. R. (2018). Payment instruments, finance and development. Journal of Development Economics, 133, 162–186. 

2.Berger, A. N., & Udell, G. F. (2002). Small business credit availability and relationship lending: The importance of bank organisational structure. Economic Journal, 112(477), 32–53.

3.Bruton, G. D., Ketchen Jr, D. J., & Ireland, R. D. (2013). Entrepreneurship as a solution to poverty. Journal of Business Venturing, 28(6), 683-689.

4.Chao, P., Biao, M., & ZHANG, C. (2021). Poverty alleviation through e-commerce: Village involvement and demonstration policies in rural China. Journal of Integrative Agriculture, 20(4), 998-1011.

5.Chinn, M. D., & Fairlie, R. W. (2010). ICT Use in the Developing World: An Analysis of Differences in Computer and Internet Penetration. Review of International Economics, 18(1), 153–167.

6.Corradin, S., & Popov, A. (2015). House prices, home equity borrowing, and entrepreneurship. Review of Financial Studies, 28(8), 2399-2428.

7.DiMaggio, P., & Bonikowski, B. (2008). Make Money Surfing the Web? The Impact of Internet Use on the Earnings of U.S. Workers. American Sociological Review, 73(2), 227–250.

8.Evans, W. N., Oates, W. E., & Schwab, R. M. (1992). Measuring peer group effects: A study of teenage behavior. Journal of Political Economy, 100(5), 966–991.

9.Galperin, H., & Viecens, F. M. (2017). Connected for development? Theory and evidence about the impact of internet technologies on poverty alleviation. Development Policy Review, 35(3), 315-336.

10.Guo, F., Wang, J.Y., Wang, F., Kong, T., Zhang, X., & Cheng, Z.Y. (2020). Measuring China's digital financial inclusion: Index compilation and spatial characteristics. China Economic Quarterly, 19(4), 1401-1418.

11.Guo, Y., Zhou, Y., & Liu, Y. (2019). Targeted poverty alleviation and its practices in rural China: A case study of Fuping county, Hebei Province. Journal of Rural Studies. https://doi.org/10.1016/j.jrurstud.2019.01.007.

12.He, X. (2019). Digital entrepreneurship solution to rural poverty: theory, practice and policy implications. Journal of Developmental Entrepreneurship, 24(1), 1950004.

13.Hsiao, K. L. (2011). Why internet users are willing to pay for social networking services. Online Information Review, 35(5), 770-788.

14.Huang, Y., & Huang, Z. (2018). The development of digital finance in China: Present and future. China Economic Quarterly, 17(1), 205-218.

15.Huang, Y., & Tao, K.(2019). Revolution of digital finance in China: Experience, impacts and implications for regulation. International Economic Review, 27(6), 24-35.

16.Imai, K. S., Arun, T., & Annim, S. K. (2010). Microfinance and Household Poverty Reduction: New Evidence from India. World Development, 38(12), 1760–1774.

17.Jack, W., Ray, A., & Suri, T. (2013). Transaction Networks: Evidence from Mobile Money in Kenya. American Economic Review, 103(3), 356–361.

18.Jack, W., & Suri, T. (2014). Risk sharing and transactions Costs: Evidence from Kenya’s mobile money revolution. American Economic Review, 104(1), 183–223.

19.James, J. (2006). The Internet and poverty in developing countries: Welfare economics versus a functionings-based approach. Futures, 38(3), 337-349.

20.Karaivanov, A. (2012). Financial constraints and occupational choice in Thai villages. Journal of Development Economics, 97(2), 201–220.

21.Kiiski, S., & Pohjola, M. (2002). Cross-country diffusion of the Internet. Information Economics and Policy, 14(2), 297–310.

22.Krueger, A. B. (1993). How Computers Have Changed the Wage Structure: Evidence from Microdata, 1984-1989. Quarterly Journal of Economics, 108(1), 33–60.

23.Lai, J. T., Yan, I. K., Yi, X., & Zhang, H. (2020). Digital financial inclusion and consumption smoothing in China. China & World Economy, 28(1), 64-93.

24.Li, J., Wu, Y., & Xiao, J. J. (2020). The impact of digital finance on household consumption: Evidence from China. Economic Modelling, 86, 317-326.

25.Liao, C., Fei, D., Huang, Q., Jiang, L., & Shi, P. (2021). Targeted poverty alleviation through photovoltaic-based intervention: Rhetoric and reality in Qinghai, China. World Development, 137, 105117.

26.Liébana-Cabanillas, F., Munoz-Leiva, F., & Sánchez-Fernández, J. (2018). A global approach to the analysis of user behavior in mobile payment systems in the new electronic environment. Service Business, 12(1), 25-64.

27.Liu, Y., Guo, Y., & Zhou, Y. (2018). Poverty alleviation in rural China: policy changes, future challenges and policy implications. China Agricultural Economic Review, 10(2), 241–259.

28.Liu, Y., Liu, J., & Zhou, Y. (2017). Spatio-temporal patterns of rural poverty in China and targeted poverty alleviation strategies. Journal of Rural Studies, 52, 66-75.

29.Montgomery, J. D. (1991). Social networks and labor-market outcomes: Toward an economic analysis. American Economic Review, 81(5), 1408-1418.

30.Mora-Rivera, J., & García-Mora, F. (2021). Internet access and poverty reduction: Evidence from rural and urban Mexico. Telecommunications Policy, 45(2), 102076.

31.Peng, C., Ma, B., & ZHANG, C. (2021). Poverty alleviation through e-commerce: Village involvement and demonstration policies in rural China. Journal of Integrative Agriculture, 20(4), 998-1011.

32.Quibria, M., Ahmed, S. N., Tschang, T., & Reyes-Macasaquit, M. L. (2003). Digital divide: determinants and policies with special reference to Asia. Journal of Asian Economics, 13(6), 811–825.

33.Shoji, M., Aoyagi, K., Kasahara, R., Sawada, Y., & Ueyama, M. (2012). Social Capital Formation and Credit Access: Evidence from Sri Lanka. World Development, 40(12), 2522–2536.

34.Si, S., Yu, X., Wu, A., Chen, S., Chen, S., & Su, Y. (2015). Entrepreneurship and poverty reduction: A case study of Yiwu, China. Asia Pacific Journal of Management, 32(1), 119–143.

35.Suri, T. (2017). Mobile Money. Annual Review of Economics, 9(1), 497–520.

36.Suri, T., & Jack, W. (2016). The long-run poverty and gender impacts of mobile money. Science, 354(6317), 1288-1292.

37.Sutter, C., Bruton, G. D., & Chen, J. (2019). Entrepreneurship as a solution to extreme poverty: A review and future research directions. Journal of Business Venturing, 34(1), 197-214.

38.Stock, J. H. , & Yogo, M. (2005). Testing for weak instruments in linear IV regression, in identification and inference for econometric models: Essay in honor of Thomas Rothenberg. Cambridge University Press.

39.Wang, H., Zhao, Q., Bai, Y., Zhang, L., & Yu, X. (2020). Poverty and subjective poverty in rural China. Social Indicators Research, 150(1), 219-242.

40.Wang, X. (2020). Mobile payment and informal business: Evidence from China's household panel data. China & World Economy, 28(3), 90-115.

41.Xie, X., Shen, X., Zhang, H., & Guo, F. (2018). Can digital fiance promote the entrepreneurship? Evidence from China. China Economic Quarterly, 17(4), 1157-1180.

42.Yin, Z., Gong, X., Guo, P., & Wu, T. (2019). What drives entrepreneurship in digital economy? Evidence from China. Economic Modelling, 82, 66-73.

43.Zhang, X., & Li, G. (2003). Does guanxi matter to nonfarm employment?. Journal of Comparative Economics, 31(2), 315-331.

44.Zhou, Y., Guo, Y., Liu, Y., Wu, W., & Li, Y. (2018). Targeted poverty alleviation and land policy innovation: Some practice and policy implications from China. Land Use Policy, 74, 53–65.

“Poverty Reduction in Rural China: Does the Digital Finance Matter?”

Response to Reviewer #2

We appreciate that our reviewer provides meaningful suggestions and comments on several details of our study, which further helps us to improve our paper. In this revised version, we attempt to address all the concerns our reviewer proposes. Our point-by-point responses to the reviewer are as follows. 

The points raised by reviewers are written in blue italics, whereas our responses are shown in normal font (single-spaced), and the key quotation of the revised manuscript is shown in red font (double-spaced). In the modified manuscript, all changes are marked in red.

1.To better clarify the significance of this research, the logic of the introduction needs to be strengthened, and the review of existing research should focus more on the research theme.

Response:

We quite agree with the reviewer's comment. Following the reviewer's suggestion, we revised the revised and improved the introduction. First, we specifically stated the history of China’s poverty reduction and related literature, especially highlighting relevant research on targeted poverty alleviation in details. 

The revised content is as follows (on pages 2-3):

…

Poverty reduction is the basis for maintaining social stability, and it has become one of the major challenges faced by developing countries in their development. China is the largest developing country in the world and once had the largest rural poor population (Liu et al., 2017). Since 1949, China has made great efforts to solve the problems of poverty, and has implemented a series of poverty reduction measures in different stages. Before 1978, the primary objective of antipoverty was to ensure basic survival of farmers, and the main measures were low-level social assistance together with mutual aid and cooperation (Guo et al., 2019). However, in 1978, according to the rural poverty standard calculated at the price level of that year, 770 million people are still in absolute poverty, accounting for 97.5% of the rural population. From 1978 to 2012, China's institutional reform had significantly relieved the poverty in rural areas, more than 700 million people in rural China overcame the problems of poverty. In 2013, the Chinese government implemented targeted poverty alleviation (TPA). TPA ensured that assistance accurately reaches poverty-stricken villages and households, and combined five approaches to eliminate poverty, which are industrial development, resettlement, ecological compensation, strengthened education and social security (Guo et al., 2019; Liao et al., 2021; Liu et al., 2018; Zhou et al., 2018). The latest report from the China's National Bureau of Statistics shows that from 2012 to 2019, the average annual reduction rate of rural poverty was as high as 51.06%, and China has solved the problem of absolute poverty in 2020. However, the relative poverty of rural households remains severe due to the large disparity between urban and rural development in China (Peng et al., 2021; Wang et al., 2020). 

…

Second, we reorganized the literature on financial poverty reduction in the second paragraph of “Introduction”, and revised relevant expressions, explaining the impact of financial development on poverty reduction as clearly as possible.

The revised content is as follows:

…

Among many poverty reduction approaches, the effectiveness of financial poverty alleviation has always been concerned. In terms of the macro-economic, financial development may shrink poverty through economic growth, urbanization, industrialization, and international trade (e.g., Akhter and Daly, 2009; Easterly, 1993; Ghosh, 2006; Greenwood and Jovanovic, 1990; Jeanneney and Kpodar, 2011; Levine et al, 2000; Rousseau and D'Onofrio, 2013; Uddin et al, 2014; Van Horen, 2007). From the micro perspective, financial development may reach more low-income groups and reduce the incidence of relative poverty, especially as countries increasingly focus on inclusive financial development (Chibba, 2009; Guo et al., 2020; Kapoor, 2014; Lai et al., 2020; Li et al., 2018; Neaime and Gaysset, 2018; Sarma and Pais, 2011). In recent years, digital finance has received widespread attention as financial development and the Internet have become more and more closely integrated.

Digital finance is a new financial format that relies on the Internet and information technology tools to carry out financial services and benefit more groups (Guo et al., 2020; Huang and Huang, 2018; Lai et al., 2020; Li et al., 2020). In essence, it is an important type and application of Financial Technology (FinTech) (Goldstein et al., 2019). China's digital finance is mainly mobile payments, online loans, digital insurance and online investments (Huang and Tao, 2019; Li et al., 2020). With the spread of the Internet and smartphones, digital finance in China has made great strides, which has greatly increased the accessibility and convenience of formal financial services, especially for those who previously did not have access to them (Liu et al., 2021; Ozili, 2018). However, since research on the impact of digital finance on poverty reduction is still very limited, we try to explore the role of digital finance in China’s rural poverty reduction, as China is the most widely used country for digital finance in the world.

The role of digital finance has been noted by many scholars. On the one hand, they found that digital finance not only promotes economic growth, but also plays a positive role in reducing the rural-urban gap (Jiang et al., 2021). On the other hand, in terms of the impact on individuals and households, the functions of digital finance can be attributed as: easing the financing constraints of low-income groups (Wang, 2020; Yin et al., 2019), achieving consumption smoothing (Lai et al., 2020; Li et al., 2020; Zhang et al., 2021), promoting the possibility of entrepreneurial activities (Wang, 2020; Xie et al., 2018), and increasing the potential benefits of entrepreneurship (Beck et al., 2018; Yin et al., 2019). Additionally, few studies explored the impact of digital finance on poverty alleviation. Another literature similar to our study comes from Suri and Jack (2016), who obtained the conclusion that FinTech contributes to poverty reduction. They found that M-Pesa, which is mobile banking service launched by mobile operator “Safaricom” in Kenya, enabled many Kenyan women to move out of subsistence farming and into small-scale enterprises to earn higher incomes by providing additional financial resources.

However, there is some controversy in the previous literature on the poverty reduction effect of FinTech. On the one hand, FinTech requires the use of the Internet or mobile devices, but some poor people may have a digital divide (Song et al., 2020), making it difficult to realize the poverty alleviation benefits of digital finance (Neaime and Gaysset, 2018). On the other hand, poverty reduction effects of FinTech may be short-term (Bateman et al., 2019), affected by the imperfection of credit and financial systems. Therefore, further exploration is still needed on whether digital finance can effectively alleviate poverty.

…

2. The analysis of the theoretical framework is unconvincing. For example, can digital finance bring information advantages to the poor? In fact, the poor do not have information advantages, and digital finance only reduces information inequality in a sense.

Response:

We sincerely appreciate our reviewer’s suggestion. Following the reviewer's comments, we have carefully revised the section 3.2, changed the title from “Information advantages” to “Information constraints”. The literature has been added. We emphasized that digital finance is a new financial format that combines the ICT with traditional financial services, and has promoted the development of financial inclusion, benefiting more low-income people. With the help of financial platforms and big data technology, digital finance can deliver information that is more useful to clients to improve their economic conditions and is more compatible with user characteristics, further alleviating the information constraints of poor people.

The revised content is as follows (on page 7):

…

3.2. Information constraints

In addition to credit constraints, poor and low-income rural households also face strong information constraints. There is a clear “digital gap” with middle-income and high-income groups in the production, employment, and life for the poor. Some studies confirmed the impact of the digital divide on the income gap (Chinn and Fairlie, 2010; Kiiski and Pohjola, 2002; Quibria et al., 2003). On the contrary, with the rapid development of information and communication technology (ICT), the use of smartphones and the Internet has a significant role in increasing individual income (Krueger, 1993; DiMaggio and Bonikowski, 2008). Digital finance is a new financial format that combines the ICT with traditional financial services to reach more groups (Lai et al., 2020). Therefore, the development of digital finance may further strengthen the role of ICT in narrowing the income gap and further promote poverty reduction by alleviating the information constraints of poor and low-income households.

Furthermore, low-income people usually lack financial knowledge and have limited ability to collect and identify data from the Internet. Therefore, although the development of ICT makes it easier to obtain information and reduces the cost of obtaining information, it may still be difficult to benefit low-income groups. With the help of financial platforms and big data technology, digital finance will deliver information that is more useful to clients to improve their economic conditions and is more compatible with user characteristics (Guo et al., 2020; Xie et al., 2018). People can easily obtain information related to agricultural production and management, employment, finance and daily life timely from digital financial platforms (Wang, 2020; Yin et al., 2019). After big data analysis, this part of information is highly matched with users, more accurate and transparent. It may help to promote the employment of rural laborers and improve the efficiency of agricultural production (Liu et al., 2021), thus increase their income and reducing the incidence of poverty. In addition, even if the information received is only about daily life, rural households have the opportunity to reallocate resources optimally and improve their ability to cope with external risk shocks (Huang and Huang, 2018). To sum up, we propose the second hypothesis: 

Hypothesis 2: Digital finance is likely to curb rural poverty by leveraging information and alleviating information constraints.

…

3. The spatial scale of DFII data is province, but the empirical analysis takes the prefecture-level city as the spatial unit. Therefore, this is questionable.

Response:

Following the reviewer's comments, we removed Table 1 and Fig 2 on the DFII at the provincial level, and modified Figure 1 in the Section 2 of "Digital Finance in China" In addition, we changed it to the analysis of prefecture-level city data, which is consistent with the empirical analysis.

The revised content is as follows (on pages 7-8):

…

According to the Digital Financial Inclusion Index (DFII) compiled by the Institute of Digital Finance of Peking University in collaboration with Ali Finance, we found some characteristics of digital finance development in China. First, as shown in Fig 1, from 2011 to 2018, digital finance has developed rapidly in China. Second, the differences in city-level DFII between regions are gradually converging in Fig 2 and the differences between regions are narrowing, which is consistent with the findings from Huang and Tao (2019). They found that the difference in DFII between the most and least developed regions of the Chinese economy has decreased from 50.4% in 2011 to 1.4% in 2018.

Fig 1. The box-plot of municipal DFII in China from 2011 to 2018

…

4. The empirical analysis of antipoverty mechanism is not deep and persuasive.

Response:

We sincerely appreciate our reviewer’s suggestion. We redesigned the empirical analysis and modified all the empirical tables with additional methods to verify the validity of these mechanisms.

The revised content is as follows (on pages 16-22):

…

5.2. Mechanisms of poverty reduction

5.2.1. Credit constraints

As noted above, digital finance provides financial accessibility to rural households and reduces their credit constraints to alleviate poverty. Credit constraint refers to a binary variable indicating whether the household applied for a loan from a bank or credit union, but was rejected. If rural households did experience this situation suggests that they faced credit constraints, the variable is set to 1, and 0 otherwise.

In column (1) of Table 4, the estimates show a significant negative association between digital finance and rural households' credit constraints, implying that an increase in the level of digital finance is effective in alleviating households' credit constraints. In columns (2) and (3), the two digital finance sub-indicators are also negative and significant at the 1% level, indicating that digital financial development reduces the likelihood that rural households experience credit constraint distress. Moreover, in columns (4) and (5), we find that credit constraints are indeed positively associated with household absolute poverty and relative poverty, which is consistent with the findings in previous studies (e.g., Bernheim et al., 2015; Morduch, 1994; Ranjan, 2001; Zhang et al., 2020).

Table 4. Digital finance, credit constraints, and rural household poverty

 (1) (2) (3) (4) (5)

 Credit constraint Absolute poverty Relative poverty 

Digital finance -0.1215*** 

 (0.0218) 

Breadth -0.0951*** 

 (0.0170) 

Depth -0.1275*** 

 (0.0228) 

Credit constraint 0.0358** 0.0725***

 (0.0151) (0.0140)

Control variables Yes Yes Yes Yes Yes

City fixed effects Yes Yes Yes Yes Yes

N 11,687 11,687 11,687 11,687 11,687

Notes: The significance levels of 1%, 5%, and 10% are denoted by ***, **, and *, respectively. Standard errors clustered at the city level are reported in parentheses. Baseline control variables and city fixed effects are added in all regressions.

Additionally, as shown in Table 1, digital finance indicator system includes some sub-indexes related to household credit constraints, such as the lending and credit in usage depth; thus, we further consider these indicators to validate the credit constraint mechanism. Table 5 presents the results. We find that both lending and credit reduce poverty among rural households and the estimated coefficients are all significant at the 1% level, suggesting that the credit function of digital finance helps alleviate poverty (Liu et al., 2021; Yin et al., 2019). All in all, these results provide supportive evidence for Hypothesis 1 and confirm that digital finance could help Chinese rural households escape poverty by easing their credit constraints.

Table 5. Digital financial indicators involving credit and rural household poverty

 (1) (2) (3) (4)

 Absolute poverty Relative poverty

Lending -0.2417*** -0.4312*** 

 (0.0587) (0.0707) 

Credit -0.0528*** -0.0941***

 (0.0128) (0.0154)

Control variables Yes Yes Yes Yes

City fixed effects Yes Yes Yes Yes

N 11,686 11,686 11,686 11,686

Notes: The significance levels of 1%, 5%, and 10% are denoted by ***, **, and *, respectively. Standard errors clustered at the city level are reported in parentheses. Baseline control variables and city fixed effects are added in all regressions.

5.2.2. Information constraints

 As discussed in Section 3, digital finance is based on the Internet and big data technology, which can help rural households alleviate their poverty by alleviating their information constraints. We construct two variables related to household information access, Information attention and Mobile payment. The former is an ordered variable from 1 to 5, using the householder's concern for economic and financial information, with larger values indicating stronger information concerns. The latter is a binary variable measured by whether rural householders use mobile payment. The reason why mobile payment is regarded as a proxy for information advantages is that mobile payments are becoming an important way for households to access financial and economic information (Wang, 2020; Yin et al., 2019). 

In the first three columns of Table 6, the estimates suggest that digital finance is positively associated with rural householders' information attentions. Similarly, in the last three columns, the coefficients on Digital finance are all positive and statistically significant, indicating that the digital finance similarly increases the probability of mobile payment use by rural households. These results indicate that digital finance increases rural people's attention to economic and financial information, raises their use of mobile payments, and create information advantages for them.

Table 6. Information constraints of digital finance

 (1) (2) (3) (4) (5) (6)

 Information attention Mobile payment 

Digital finance 0.4021*** 0.0565** 

 (0.0774) (0.0284) 

Breadth 0.3146*** 0.0442** 

 (0.0606) (0.0222) 

Depth 0.4219*** 0.0593**

 (0.0812) (0.0298)

Control variables Yes Yes Yes Yes Yes Yes

City fixed effects Yes Yes Yes Yes Yes Yes

N 11,786 11,786 11,786 11,816 11,816 11,816

Notes: The significance levels of 1%, 5%, and 10% are denoted by ***, **, and *, respectively. Standard errors clustered at the city level are reported in parentheses. Baseline control variables and city fixed effects are added in all regressions.

As before, we further tested whether these two mechanisms could reduce absolute and relative poverty among rural households, and the results are shown in Table 7. It is clear that all estimated coefficients on Information attention and Mobile payment are significantly negative, which remains consistent with some literature (Mora-Rivera and García-Mora, 2021; James, 2006). These findings provide a preliminary indication for the reliability of hypothesis 2, that the information advantage from digital finance helps to alleviate rural household poverty.

Table 7. Information constraints and rural household poverty

 (1) (2) (3) (4)

 Absolute poverty Relative poverty

Information attention -0.0370*** -0.0247*** 

 (0.0033) (0.0039) 

Mobile payment -0.0173** -0.0341***

 (0.0075) (0.0122)

Control variables Yes Yes Yes Yes

City fixed effects Yes Yes Yes Yes

N 11,786 11,816 11,786 11,816

Notes: The significance levels of 1%, 5%, and 10% are denoted by ***, **, and *, respectively. Standard errors clustered at the city level are reported in parentheses. Baseline control variables and city fixed effects are added in all regressions.

Furthermore, since the Internet is the most dominant information exchange platform (Galperin and Viecens, 2017; Hsiao, 2011), and digital finance is also used to realize various financial services through the Internet (Guo et al., 2020; Huang and Huang, 2018; Li et al., 2020), we further introduce a moderator variable, Internet use, and construct and interaction term to fully verify the information advantage characteristics of digital finance. In table 8, the estimates show that although the coefficients on interaction terms are negative in the first three columns, they are insignificant. In contrast, in the last three columns, the interaction terms for digital finance and Internet use are all significantly negative, suggesting that digital finance can achieve a reduction in relative poverty among rural households through the information channel of the Internet. Taken together, by using a variety of methods, we support the hypothesis 2 that digital finance is likely to reduce poverty by alleviating information constraints of rural households.

Table 8. Digital finance, Internet use, and rural household poverty

 (1) (2) (3) (4) (5) (6)

 Information attention Mobile payment 

Digital finance -0.1144*** -0.2264*** 

 (0.0276) (0.0326) 

Digital finance*Internet use -0.0033 -0.0167*** 

 (0.0034) (0.0047) 

Breadth -0.0880*** -0.1754*** 

 (0.0213) (0.0251) 

Breadth*Internet use -0.0030 -0.0191*** 

 (0.0036) (0.0052) 

Depth -0.1203*** -0.2383***

 (0.0291) (0.0344)

Depth*Internet use -0.0030 -0.0151***

 (0.0031) (0.0043)

Control variables Yes Yes Yes Yes Yes Yes

City fixed effects Yes Yes Yes Yes Yes Yes

N 11,743 11,743 11,743 11,743 11,743 11,743

Notes: The significance levels of 1%, 5%, and 10% are denoted by ***, **, and *, respectively. Standard errors clustered at the city level are reported in parentheses. Baseline control variables and city fixed effects are added in all regressions.

5.2.3. Social networks

In Hypothesis 3, we consider that another important mechanism for poverty reduction effect of digital finance is to help expand the social networks of rural households. Given the complexity of social network measurement, several previous studies used money gift income and expenditures and the spending on social network maintenance as proxies for household social networks (Hudik and Fang, 2020; Zhang and Li, 2003). The CHFS provides two types of variables in terms of income and expenditure associated with social networks. For social network income, we select two variables, Money gift receive (dummy) and Money gift incomes; for social network expenditure, Money gift expenditure (dummy) and Maintenance expenditure[ Maintenance expenses related to social networks include transportation expenses, recreation expenses, and communication expenses in 1000 yuan.] were selected as mechanism variables.

Table 9 examines the effects of digital finance on households' social networks from the perspective of income. The estimates in columns (1)-(3) show that there is no association between digital finance and money gift receive of rural households. However, in columns (4)-(6) of Table 9, we find that coefficients on Digital finance are all positive and significant at the 1% level, implying that the digital finance leads to an increase in money gifts received by rural households. In the last two columns, not surprisingly, the estimates indicate that gift income, as a liquid monetary asset, helps rural households escape poverty.

Table 9. Digital finance, social network, and rural household poverty (revenue related to social networks)

 (1) (2) (3) (4) (5) (6) (7) (8)

 Money gift receive Money gift incomes Absolute poverty Relative poverty

Digital finance 0.0084 5.9673*** 

 (0.0374) (0.2265) 

Breadth 0.0065 9.7702*** 

 (0.0293) (0.3708) 

Depth 0.0088 2.8391*** 

 (0.0392) (0.1078) 

Money gift incomes -0.0310*** -0.0304***

 (0.0040) (0.0043)

Control variables Yes Yes Yes Yes Yes Yes Yes Yes

City fixed effects Yes Yes Yes Yes Yes Yes Yes Yes

N 11,773 11,773 11,773 5236 5236 5236 5236 5236

Notes: The significance levels of 1%, 5%, and 10% are denoted by ***, **, and *, respectively. Standard errors clustered at the city level are reported in parentheses. Baseline control variables and city fixed effects are added in all regressions.

Moreover, from a social network spending perspective, we further explore whether digital finance can alleviate poverty through social networks. As reported in Table 10, although digital finance significantly reduces the probability of rural households spending on money gifts in columns (1)-(3), it leads to an increase in household spending related to maintaining social networks in the last three columns. Further, in Table 11, we find that money gift expenditure are positively associated with rural household poverty, while there is no association between maintenance expenditure and rural household poverty. These results suggest that while digital finance helps rural households expand their social networks, the additional expenditures incurred may not be conducive to lifting poor rural households out of poverty. Therefore, our findings only partially support Hypothesis 3. However, considering that our measure cannot fully capture all dimensions of social networks of rural households, our estimates provide only suggestive evidence.

Table 10. Digital finance and expenses related to social networks

 (1) (2) (3) (4) (5) (6)

 Money gift expenditure Maintenance expenditure

Digital finance -1.2337*** 4.3323*** 

 (0.0305) (1.0130) 

Breadth -0.9651*** 3.3893*** 

 (0.0239) (0.7925) 

Depth -1.2942*** 4.5449***

 (0.0320) (1.0627)

Control variables Yes Yes Yes Yes Yes Yes

City fixed effects Yes Yes Yes Yes Yes Yes

N 11,786 11,786 11,786 11,816 11,816 11,816

Notes: The significance levels of 1%, 5%, and 10% are denoted by ***, **, and *, respectively. Standard errors clustered at the city level are reported in parentheses. Baseline control variables and city fixed effects are added in all regressions.

Table 11. Expenses related to social networks and rural household poverty

 (1) (2) (3) (4)

 Absolute poverty Relative poverty

Money gift expenditure 0.0520*** 0.0537*** 

 (0.0090) (0.0091) 

Maintenance expenditure 0.0000 0.0002

 (0.0002) (0.0002)

Control variables Yes Yes Yes Yes

City fixed effects Yes Yes Yes Yes

N 11,786 11,816 11,786 11,816

Notes: The significance levels of 1%, 5%, and 10% are denoted by ***, **, and *, respectively. Standard errors clustered at the city level are reported in parentheses. Baseline control variables and city fixed effects are added in all regressions.

5.2.4. Entrepreneurial activities

As highlighted in Section 3, another explanation for digital finance to alleviate rural household poverty is entrepreneurial activities. We choose two binary variables, namely entrepreneurship and online sale. With the advent of the Internet economy, online sale as a form of informal entrepreneurship has also become popular among Chinese families (Yin et al., 2019).

In Table 12, we further explore the impact of digital finance on rural households' entrepreneurial activities to test Hypothesis 4. The estimates show that, as expected, digital finance significantly increases rural households' likelihood of entrepreneurship in the first three columns. In addition, the coefficients on Digital finance are insignificant in columns (4)-(6), indicating that digital finance does not increase the probability of rural households selling online. The possible reason is that online sales need a good logistics base. Compared to urban areas, the logistics system in rural China is still lagging behind, which could also hinder the stimulating effect of digital finance on online sales. 

Additionally, in columns (7) and (8) of Table 12, The coefficient on Entrepreneurship is significantly negative, which indicates that entrepreneurship help rural households to escape from poverty, as emphasized by some previous research (e.g., Bruton et al., 2013; Ghani et al., 2014; Sutter et al., 2019). In summary, these estimates support our theoretical expectations in Hypothesis 4 and suggest that digital finance may reduce rural household poverty primarily through offline entrepreneurship.

Table 12. Digital finance, entrepreneurship, and rural household poverty

 (1) (2) (3) (4) (5) (6) (7) (8)

 Entrepreneurship Online sale Absolute poverty Relative poverty

Digital finance 0.2062*** 0.0084 

 (0.0318) (0.0101) 

Breadth 0.1613*** 0.0066 

 (0.0249) (0.0079) 

Depth 0.2164*** 0.0088 

 (0.0334) (0.0106) 

Entrepreneurship -0.0212** -0.0285***

 (0.0085) (0.0105)

Control variables Yes Yes Yes Yes Yes Yes Yes Yes

City fixed effects Yes Yes Yes Yes Yes Yes Yes Yes

N 11,816 11,816 11,816 11,743 11,743 11,743 11,816 11,816

Notes: The significance levels of 1%, 5%, and 10% are denoted by ***, **, and *, respectively. Standard errors clustered at the city level are reported in parentheses. Baseline control variables and city fixed effects are added in all regressions.

…

5. “5.4 Robustness checks”: the logarithm of the distance to Hangzhou? This analysis is not credible. Please reconsider relevant content.

Response:

We sincerely appreciate our reviewer’s suggestion. We replaced the instrumental variable (IV) in the revised manuscript. Referring to previous studies (Li et al., 2020; Xie et al., 2018), we use provincial historical Internet penetration as an IV. 

The revised content is as follows (on pages 25-27):

…

5.4.1. IV methods

Although we control for city fixed effects and cluster at the city level, some potential endogeneity problems could not be completely ruled out. Therefore, we adopt the IV methods to perform robustness tests. Referring to previous studies (Li et al., 2020; Xie et al., 2018), we use provincial Internet penetration as an IV, and the original data were obtained from the Statistical Report on the Internet Development in China. 

A good instrumental variable needs to satisfy both relevance assumption and exclusion restriction assumption. From the perspective of relevance assumption, the diffusion and popularity of the Internet is an important basic condition for the development of digital finance (Liu et al., 2021; Xie et al., 2020), and digital finance tends to grow better in regions with better Internet infrastructure in China (Guo et al., 2020; Huang and Tao, 2019). Therefore, Internet penetration and digital finance development are closely linked. In terms of the exclusion restriction hypothesis, considering that some previous studies concluded the role of Internet infrastructure in poverty alleviation (e.g., Chao et al., 2021; Galperin and Viecens, 2017; James, 2006; Mora-Rivera and García-Mora, 2021), we use historical Internet penetration as an IV[ Since the earliest data provided by the Statistical Report on Internet Development in China is 1997, we use the provincial Internet penetration in 1997 as the IV. ]. After controlling for the city fixed effects, it is difficult for historical provincial Internet penetration to directly affect household poverty through other channels, which makes our selected IV theoretically feasible.

We employ the two stage least square (2SLS) method, and the results of the first stage are shown in Table 16. We find the IV, historical Internet penetration, is positively correlated with Digital finance, with statistical significance at the 1% level. More importantly, the first-stage F value in the first two columns is well above the Stock-Yogo critical value for a weak IV (Stock and Yogo, 2005)[ In column (3), the first-stage F value less than 10. In the second-stage results, the Anderson-Rubin Wald test suggests that our IV is strong (the P-value is less than 0.05).]. In summary, the first-stage estimated results indicate that historical Internet penetration contributes to the digital finance development in China.

Table 16. The impact of digital finance on rural household poverty: IV methods (first-stage results)

 (1) (2) (3)

 Digital finance Breadth Depth

Historical Internet penetration 0.3014*** 0.3620*** 0.3650***

 (0.0832) (0.0843) (0.1315)

Control variables Yes Yes Yes

City fixed effects Yes Yes Yes

First-stage F value 13.1176 18.4637 7.7104

N 11,816 11,816 11,816

Notes: The significance levels of 1%, 5%, and 10% are denoted by ***, **, and *, respectively. Standard errors clustered at the city level are reported in parentheses. Baseline control variables and city fixed effects are added in all regressions.

Table 17 shows the second stage results. Not surprisingly, all the coefficients of the variables related to digital finance are significantly negative at the 1%level. Based on columns (1) and (4), the IV estimates suggest that for each unit increase in the digital finance aggregation index, the probability of absolute poverty and relative poverty among rural households decreases by 9.5% and 16.84%, respectively, which is quite close to the OLS estimates in Table 3. Thus, the IV estimates suggest that our main specification is robust and digital finance does play an important role in reducing poverty in rural China.

Table 17. The impact of digital finance on rural household poverty: IV methods (second-stage results)

 (1) (2) (3) (4) (5) (6)

 Absolute poverty Relative poverty

Digital finance -0.0950*** -0.1684*** 

 (0.0239) (0.0287) 

Breadth -0.0744*** -0.1318*** 

 (0.0187) (0.0225) 

Depth -0.0997*** -0.1767***

 (0.0250) (0.0302)

Control variables Yes Yes Yes Yes Yes Yes

City fixed effects Yes Yes Yes Yes Yes Yes

Anderson-Rubin Wald test 4.2934 4.2934 4.2934 42.0415 42.0415 42.0415

P-value 0.0383 0.0383 0.0383 0.0000 0.0000 0.0000

N 11,816 11,816 11,816 11,816 11,816 11,816

Notes: The significance levels of 1%, 5%, and 10% are denoted by ***, **, and *, respectively. Standard errors clustered at the city level are reported in parentheses. Baseline control variables and city fixed effects are added in all regressions.

…

6. In the past, a large number of the poor in rural China were old, weak, sick and disabled, but they were excluded in this study. This makes the results questionable.

Response:

In the initial version, considering the balance of the samples we removed these special samples for robustness testing. We strongly agree with the reviewer's suggestion, so in the revised version we removed these robustness tests in Table 19.

7. The policy implications is not targeted and needs to be strengthened. For example, digital financial infrastructure is seriously insufficient in less developed countries, and their first problem is to promote the construction of digital financial infrastructure. However, the research only outlines the need to strengthen digital finance, but did not analyze how to achieve it. Therefore, the policy enlightenment is unrealistic.

Response:

We quite agree with our reviewer’s suggestion that we should strengthen the policy implications. The revised policy implications emphasizes that China should further promote the construction of digital financial infrastructure in underdeveloped regions, through government financial support and guidance of the related policy. In addition, we propose that government’s poverty alleviation department can cooperate with research institutions and digital financial institutions through the establishment of poverty alleviation funds, to improve the digital financial services to benefit more disadvantaged groups.

The revised content is as follows (on pages 19-20):

…

The relevant policy implications are very clear. First, our results indicate that digital finance has a significant effect on the alleviation of relative poverty. Therefore, Chinese government should further promote the construction of digital financial infrastructure in underdeveloped regions through government financial support and guidance of the related policy, such as increasing smartphone penetration, accelerating the construction of 5G networks and the application of big data technologies, and enable digital finance to benefit more low-income and poor groups. Second, our findings suggest that digital finance does not appear to be sufficient in alleviating the relative poverty of some older and uneducated people. The government’s poverty alleviation department proposes to establish some cooperative projects with research institutions and digital financial institutions to investigate the difficulties and needs of the elderly and low-educated people in using digital financial services, and further improve the platform, which is more beneficial to disadvantaged groups. 

…

References

1.Akhter, S., & Daly, K. J. (2009). Finance and poverty: Evidence from fixed effect vector decomposition. Emerging Markets Review, 10(3), 191–206. 

2.Beck, T., Pamuk, H., Ramrattan, R., & Uras, B. R. (2018). Payment instruments, finance and development. Journal of Development Economics, 133, 162–186. 

3.Chao, P., Biao, M., & ZHANG, C. (2021). Poverty alleviation through e-commerce: Village involvement and demonstration policies in rural China. Journal of Integrative Agriculture, 20(4), 998-1011.

4.Chibba, M. (2009). Financial inclusion, poverty reduction and the millennium development goals. European Journal of Development Research, 21(2), 213–230. 

5.Chinn, M. D., & Fairlie, R. W. (2010). ICT Use in the Developing World: An Analysis of Differences in Computer and Internet Penetration. Review of International Economics, 18(1), 153–167.

6.Corradin, S., & Popov, A. (2015). House prices, home equity borrowing, and entrepreneurship. Review of Financial Studies, 28(8), 2399-2428.

7.DiMaggio, P., & Bonikowski, B. (2008). Make Money Surfing the Web? The Impact of Internet Use on the Earnings of U.S. Workers. American Sociological Review, 73(2), 227–250.

8.Easterly, W. (1993). How much do distortions affect growth? Journal of Monetary Economics, 32(2), 187–212.

9.Galperin, H., & Viecens, F. M. (2017). Connected for development? Theory and evidence about the impact of internet technologies on poverty alleviation. Development Policy Review, 35(3), 315-336.

10.Ghosh, S. (2006). Did financial liberalization ease financing constraints? Evidence from Indian firm-level data. Emerging Markets Review, 7(2), 176–190. 

11.Goldstein, I., Jiang, W., & Karolyi, G. A. (2019). To FinTech and beyond. Review of Financial Studies, 32(5), 1647–1661.

12.Greenwood, J., & Jovanovic, B. (1990). Financial development, growth, and the distribution of income. Journal of Political Economy, 98(5), 1076–1107.

13.Guo, F., Wang, J.Y., Wang, F., Kong, T., Zhang, X., & Cheng, Z.Y. (2020). Measuring China's digital financial inclusion: Index compilation and spatial characteristics. China Economic Quarterly, 19(4), 1401-1418.

14.Guo, Y., Zhou, Y., & Liu, Y. (2019). Targeted poverty alleviation and its practices in rural China: A case study of Fuping county, Hebei Province. Journal of Rural Studies. https://doi.org/10.1016/j.jrurstud.2019.01.007.

15.Hsiao, K. L. (2011). Why internet users are willing to pay for social networking services. Online Information Review, 35(5), 770-788.

16.Huang, Y., & Huang, Z. (2018). The development of digital finance in China: Present and future. China Economic Quarterly, 17(1), 205-218.

17.Huang, Y., & Tao, K.(2019). Revolution of digital finance in China: Experience, impacts and implications for regulation. International Economic Review, 27(6), 24-35.

18.Hudik, M., & Fang, E. S. (2020). Money or in-kind gift? Evidence from red packets in China. Journal of Institutional Economics, 16(5), 731-746.

19.Jack, W., & Suri, T. (2014). Risk sharing and transactions Costs: Evidence from Kenya’s mobile money revolution. American Economic Review, 104(1), 183–223.

20.James, J. (2006). The Internet and poverty in developing countries: Welfare economics versus a functionings-based approach. Futures, 38(3), 337-349.

21.Jeanneney, S. G., & Kpodar, K. (2011). Financial Development and Poverty Reduction: Can There be a Benefit without a Cost? Journal of Development Studies, 47(1), 143–163.

22.Jiang, X., Wang, X., Ren, J., & Xie, Z. (2021). The Nexus between Digital Finance and Economic Development: Evidence from China. Sustainability, 13(13), 7289.

23.Kapoor, A. (2014). Financial inclusion and the future of the Indian economy. Futures, 56, 35–42. 

24.Kiiski, S., & Pohjola, M. (2002). Cross-country diffusion of the Internet. Information Economics and Policy, 14(2), 297–310.

25.Krueger, A. B. (1993). How Computers Have Changed the Wage Structure: Evidence from Microdata, 1984-1989. Quarterly Journal of Economics, 108(1), 33–60.

26.Lai, J. T., Yan, I. K., Yi, X., & Zhang, H. (2020). Digital financial inclusion and consumption smoothing in China. China & World Economy, 28(1), 64-93.

27.Levine, R., Loayza, N., & Beck, T. (2000). Financial intermediation and growth: Causality and causes. Journal of Monetary Economics, 46(1), 31–77.

28.Li, J., Wu, Y., & Xiao, J. J. (2020). The impact of digital finance on household consumption: Evidence from China. Economic Modelling, 86, 317-326.

29.Liao, C., Fei, D., Huang, Q., Jiang, L., & Shi, P. (2021). Targeted poverty alleviation through photovoltaic-based intervention: Rhetoric and reality in Qinghai, China. World Development, 137, 105117.

30.Liu, Y., Guo, Y., & Zhou, Y. (2018). Poverty alleviation in rural China: policy changes, future challenges and policy implications. China Agricultural Economic Review, 10(2), 241–259.

31.Liu, Y., Liu, J., & Zhou, Y. (2017). Spatio-temporal patterns of rural poverty in China and targeted poverty alleviation strategies. Journal of Rural Studies, 52, 66-75.

32.Morduch, J. (1994). Poverty and vulnerability. American Economic Review, 84(2), 221-225.

33.Mora-Rivera, J., & García-Mora, F. (2021). Internet access and poverty reduction: Evidence from rural and urban Mexico. Telecommunications Policy, 45(2), 102076.

34.Neaime, S., & Gaysset, I. (2018). Financial inclusion and stability in MENA: Evidence from poverty and inequality. Finance Research Letters, 24, 230–237.

35.Ozili, P. K. (2018). Impact of digital finance on financial inclusion and stability. Borsa Istanbul Review, 18(4), 329-340.

36.Peng, C., Ma, B., & ZHANG, C. (2021). Poverty alleviation through e-commerce: Village involvement and demonstration policies in rural China. Journal of Integrative Agriculture, 20(4), 998-1011.

37.Quibria, M., Ahmed, S. N., Tschang, T., & Reyes-Macasaquit, M. L. (2003). Digital divide: determinants and policies with special reference to Asia. Journal of Asian Economics, 13(6), 811–825.

38.Ranjan, P. (2001). Credit constraints and the phenomenon of child labor. Journal of Development Economics, 64(1), 81-102.

39.Rousseau, P. L., & D’Onofrio, A. (2013). Monetization, financial development, and growth: Time series evidence from 22 countries in Sub-Saharan Africa. World Development, 51, 132–153. 

40.Sarma, M., & Pais, J. (2011). Financial inclusion and development. Journal of International Development, 23(5), 613-628.

41.Shoji, M., Aoyagi, K., Kasahara, R., Sawada, Y., & Ueyama, M. (2012). Social Capital Formation and Credit Access: Evidence from Sri Lanka. World Development, 40(12), 2522–2536.

42.Song, Z., Wang, C., & Bergmann, L. (2020). China’s prefectural digital divide: Spatial analysis and multivariate determinants of ICT diffusion. International Journal of Information Management, 52, 102072.

43.Stock, J. H. , & Yogo, M. (2005). Testing for weak instruments in linear IV regression, in identification and inference for econometric models: Essay in honor of Thomas Rothenberg. Cambridge University Press.

44.Uddin, G. S., Shahbaz, M., Arouri, M., & Teulon, F. (2014). Financial development and poverty reduction nexus: A cointegration and causality analysis in Bangladesh. Economic Modelling, 36, 405–412.

45.Wang, H., Zhao, Q., Bai, Y., Zhang, L., & Yu, X. (2020). Poverty and subjective poverty in rural China. Social Indicators Research, 150(1), 219-242.

46.Wang, X. (2020). Mobile payment and informal business: Evidence from China's household panel data. China & World Economy, 28(3), 90-115.

47.Xie, X., Shen, X., Zhang, H., & Guo, F. (2018). Can digital fiance promote the entrepreneurship? Evidence from China. China Economic Quarterly, 17(4), 1157-1180.

48.Yin, Z., Gong, X., Guo, P., & Wu, T. (2019). What drives entrepreneurship in digital economy? Evidence from China. Economic Modelling, 82, 66-73.

49.Zhang, X., Yang, T., Wang, C., & Wan, G. (2020). Digital finance and household consumption: Theory and evidence from China. Management World, 36(11), 48-62.

50.Zhang, X., & Li, G. (2003). Does guanxi matter to nonfarm employment?. Journal of Comparative Economics, 31(2), 315-331.

51.Zhou, Y., Guo, Y., Liu, Y., Wu, W., & Li, Y. (2018). Targeted poverty alleviation and land policy innovation: Some practice and policy implications from China. Land Use Policy, 74, 53–65.

---

## [Decision Letter · Decision Letter 1]

29 Nov 2021

Poverty Reduction in Rural China: Does the Digital Finance Matter?

PONE-D-21-15370R1

Dear Dr. Zhao,

We’re pleased to inform you that your manuscript has been judged scientifically suitable for publication and will be formally accepted for publication once it meets all outstanding technical requirements.

Kind regards,

Mingxing Chen, Ph.D.

Academic Editor

PLOS ONE

Additional Editor Comments (optional):

Reviewers' comments:

Reviewer's Responses to Questions

**Comments to the Author**

1. If the authors have adequately addressed your comments raised in a previous round of review and you feel that this manuscript is now acceptable for publication, you may indicate that here to bypass the “Comments to the Author” section, enter your conflict of interest statement in the “Confidential to Editor” section, and submit your "Accept" recommendation.

Reviewer #1: All comments have been addressed

Reviewer #2: All comments have been addressed

2. Is the manuscript technically sound, and do the data support the conclusions?

Reviewer #1: Yes

Reviewer #2: Yes

3. Has the statistical analysis been performed appropriately and rigorously? 

Reviewer #1: Yes

Reviewer #2: Yes

4. Have the authors made all data underlying the findings in their manuscript fully available?

Reviewer #1: Yes

Reviewer #2: Yes

5. Is the manuscript presented in an intelligible fashion and written in standard English?

Reviewer #1: Yes

Reviewer #2: Yes

6. Review Comments to the Author

Reviewer #1: (No Response)

Reviewer #2: All comments have be addressd, and the manuscript has been greatly improved. Therefore, it can be accepted.

7. PLOS authors have the option to publish the peer review history of their article (what does this mean?). If published, this will include your full peer review and any attached files.

Reviewer #1: No

Reviewer #2: No

---

## [Editor Report · Acceptance letter]

3 Dec 2021

PONE-D-21-15370R1 

Poverty Reduction in Rural China: Does the Digital Finance Matter? 

Dear Dr. Zhao:

I'm pleased to inform you that your manuscript has been deemed suitable for publication in PLOS ONE. Congratulations! Your manuscript is now with our production department. 

Kind regards, 

on behalf of

Prof. Mingxing Chen 

Academic Editor

PLOS ONE